# Black-box Coreset Variational Inference

**Dionysis Manousakas**[*]
Meta[†]
dm754@cantab.ac.uk

**Hippolyt Ritter**[*]
Meta
hippolyt@meta.com

**Theofanis Karaletsos**
Insitro[‡]
theofanis@karaletsos.com

## Abstract

Recent advances in coreset methods have shown that a selection of representative datapoints can replace massive volumes of data for Bayesian inference, preserving the relevant statistical information and significantly accelerating subsequent downstream tasks. Existing variational coreset constructions rely on either selecting subsets of the observed datapoints, or jointly performing approximate inference and optimizing pseudodata in the observed space akin to inducing points methods in Gaussian Processes. So far, both approaches are limited by complexities in evaluating their objectives for general purpose models, and require generating samples from a typically intractable posterior over the coreset throughout inference and testing. In this work, we present a black-box variational inference framework for coresets that overcomes these constraints and enables principled application of variational coresets to intractable models, such as Bayesian neural networks. We apply our techniques to supervised learning problems, and compare them with existing approaches in the literature for data summarization and inference.

## 1 Introduction

Machine learning models are widely trained using mini-batches of data [8, 22], enabling practitioners to leverage large scale data for increasingly accurate models. However, repeatedly processing these accumulating amounts of data becomes more and more hardware-intensive over time and therefore costly. Thus, efficient techniques for extracting and representing the relevant information of a dataset for a given model are urgently needed.

Recent work on coresets for probabilistic models has demonstrated that Bayesian posteriors on large scale datasets can be sparsely represented via surrogate densities defined through a weighted subset of the training data (Sparse Variational Inference; **Sparse VI**) [10] or a set of learnable weighted pseudo-observations in data space (Pseudodata Sparse Variational Inference; **PSVI**) [31]. Similar to inducing points in Gaussian Processes [50], these target to parsimoniously represent the sufficient statistics of the observed dataset that are required for inference in the model. Replacing the original data via the reduced set of (pseudo-) observations unlocks the potential of scaling up downstream learning tasks, compressing for efficient storage and visualisation, and accelerating model exploration.

Unfortunately, Sparse VI and PSVI rely on access to exact samples from the model posterior and closed form gradients of it to be evaluated and updated. Yet, the posterior for most common probabilistic models is intractable, limiting their use to the class of tractable models or necessitating the use of heuristics deviating from the core objective. Dealing with intractable models typically necessitates the use of approximate methods such as variational inference, which has been increasingly popular with the development of black-box techniques for arbitrary models [43].

In this work, we overcome these challenges and introduce a principled framework for performing approximate inference with variational families based on weighted coreset posteriors. This leads

---

[*]Equal contribution [†]Now at Amazon [‡]Research supporting this publication done at Meta

36th Conference on Neural Information Processing Systems (NeurIPS 2022).

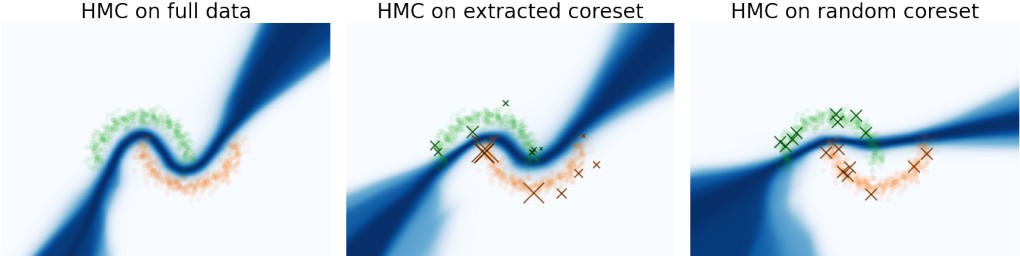

| HMC on full data | HMC on extracted coreset | HMC on random coreset |

Figure 1: Posterior inference results via Hamiltonian Monte Carlo [35] on the full training dataset, a 16-point coreset learned via BB PSVI, and a 16-point coreset comprised of randomly selected, uniformly weighted data points for half-moon binary classification with a feedforward neural network. Coreset locations are marked in crosses with the size indicating the relative weight. Regions of high predictive uncertainty are visualized by dark blue shaded areas.

to extensions of Sparse VI and PSVI to black-box models, termed **BB Sparse VI** and **BB PSVI**, respectively. As illustrated in Fig. 11 (and further detailed in the supplement) on a synthetic binary classification problem with a Bayesian neural network — an intractable model-class for prior approaches such as PSVI relying on exact posterior samples — BB PSVI is able to learn pseudodata locations and weights inducing a posterior that more faithfully represents the statistical information in the original dataset compared to random selection. However, we emphasize that the utility of coresets goes beyond applying expensive but accurate inference methods to large-scale datasets, and may drive progress in fields such as continual learning as we demonstrate later on.

Specifically, we make the following contributions in this work:

1. We derive a principled interpretation of coresets as rich variational families,

2. we modify the objective to obtain a black-box version of it using importance sampling and a nested variational inference step,

3. utilizing this objective, we propose BB PSVI and BB Sparse VI, two novel suites of variational inference algorithms for black-box probabilistic models,

4. we empirically compare to prior Bayesian and non-Bayesian coreset techniques, and

5. we showcase these techniques on previously infeasible models, Bayesian neural networks.

## 2 Background

### 2.1 Variational Inference

Consider a modeling problem where we are given a standard model:

$$p(\boldsymbol{x}, \boldsymbol{y}, \boldsymbol{\theta}) = p(\boldsymbol{y}|\boldsymbol{x}, \boldsymbol{\theta})p(\boldsymbol{\theta}). \tag{1}$$

Given a dataset $\mathcal{D} = \{(\boldsymbol{x}, \boldsymbol{y})\}$, the task of inference and maximization of the marginal likelihood $p(\boldsymbol{y}|\boldsymbol{x})$ entails estimation of the posterior distribution over model parameters $\boldsymbol{\theta}$, $p(\boldsymbol{\theta}|\boldsymbol{x}, \boldsymbol{y})$. In general, the posterior $p(\boldsymbol{\theta}|\boldsymbol{x}, \boldsymbol{y})$ is not tractable. A common solution is to resort to approximate inference, for instance via variational inference (VI). In VI, we assume that we can approximate the true posterior within a *variational family* $q(\boldsymbol{\theta}; \boldsymbol{\lambda})$ with free variational parameters $\boldsymbol{\lambda}$, by minimizing the Kullback-Leibler (KL) divergence between the approximate and the true posterior $\mathrm{D}_{\mathrm{KL}}\left(q(\boldsymbol{\theta}; \boldsymbol{\lambda})||p(\boldsymbol{\theta}|\boldsymbol{x}, \boldsymbol{y})\right)$, or equivalently by maximizing the Evidence Lower Bound (ELBO):

$$\log p(\boldsymbol{y}|\boldsymbol{x}) \geq \mathrm{ELBO}(\boldsymbol{\lambda}) = \mathbb{E}_{q(\boldsymbol{\theta}; \boldsymbol{\lambda})}\left[\log \frac{p(\boldsymbol{y}|\boldsymbol{x}, \boldsymbol{\theta})p(\boldsymbol{\theta})}{q(\boldsymbol{\theta}; \boldsymbol{\lambda})}\right]. \tag{2}$$

### 2.2 (Bayesian) coresets and pseudocoresets

Bayesian coreset constructions aim to provide a highly automated, data austere approach to the problem of inference, given a dataset $\mathcal{D}$ and a probabilistic model Eq. (1). Agnostic to the particular

inference method used, Bayesian coresets aim to construct a dataset smaller than the original one such that the posterior of the model of interest given the coreset closely matches the posterior over the original dataset, i.e. $p(\boldsymbol{\theta}|\boldsymbol{u}, \boldsymbol{z}, \boldsymbol{v}) \approx p(\boldsymbol{\theta}|\boldsymbol{x}, \boldsymbol{y})$. The coreset is a weighted dataset representation $(\boldsymbol{v}, \boldsymbol{u}, \boldsymbol{z})$ , where $\boldsymbol{v}$ denote a set of positive weights per datapoint, $||\boldsymbol{v}||_0 = M \ll N$ and $\{\boldsymbol{u}_m, \boldsymbol{z}_m\}$ are the input and output vectors for the coreset. Initial Bayesian coreset constructions formulated the problem as yielding a sparsity-constrained approximation of the true data likelihood function uniformly over the parameter space. This perspective has admitted solutions via convexifications relying on importance sampling and conditional gradient methods [12, 23], greedy methods for geodesic ascent [11], as well as non-convex approaches employing iterative hard thresholding [56]. Recently, Sparse VI [10] reformulated the problem as performing directly VI within a sparse exponential family of approximate posteriors defined on the coreset datapoints.

**PSVI** The idea of using a set of learnable weighted *pseudodata* (or *inducing points*) $(\boldsymbol{v}, \boldsymbol{u}, \boldsymbol{z}) := \{(\boldsymbol{v}_m, \boldsymbol{u}_m, \boldsymbol{z}_m)_{m=1}^M\}$ to summarize a dataset for the purpose of posterior inference was introduced in [31], as the following objective:

$$\boldsymbol{v}^\star, \boldsymbol{u}^\star, \boldsymbol{z}^\star = \underset{\boldsymbol{v}, \boldsymbol{u}, \boldsymbol{z}}{\arg\min} \, \mathrm{D}_{\mathrm{KL}} \left( p(\boldsymbol{\theta}|\boldsymbol{u}, \boldsymbol{z}, \boldsymbol{v}) || p(\boldsymbol{\theta}|\boldsymbol{x}, \boldsymbol{y}) \right). \tag{3}$$

This objective facilitates learning weighted pseudodata such that the exact posterior of model parameters given pseudodata $p(\boldsymbol{\theta}|\boldsymbol{u}, \boldsymbol{z}, \boldsymbol{v})$ approximates the exact posterior given the true dataset $p(\boldsymbol{\theta}|\boldsymbol{x}, \boldsymbol{y})$, and was demonstrated on a variety of models as a generally applicable objective. However, in [31] it was assumed that this objective is tractable, as demonstrated in experiments on models admitting analytical posteriors, or via the use of heuristics to sample from the coreset posterior, as is the case in a logistic regression employing a Laplace approximation.

**Sparse VI** Sparse VI [10] targets to incrementally minimize the rhs of Eq. (3) via selecting a subset of the dataset as a coreset, thereby fixing the locations to observed data, and only optimizing their weights $\boldsymbol{v}$. In its original construction, the coreset points are initialized to the empty set and $\boldsymbol{v}$ to $\boldsymbol{0}$. In the general case of an intractable model, the coreset posterior is computed via Monte Carlo (and in the code optionally via Laplace approximation) on the weighted datapoints of the coreset $(\boldsymbol{v}, \boldsymbol{u}, \boldsymbol{z})$. The next point selection step involves: (*i*) computing approximations of the centered log-likelihood function of each datapoint $(\boldsymbol{x}_n, \boldsymbol{y}_n)$ for model parameters $\boldsymbol{\theta}_s$ sampled from the current coreset posterior, via an MC estimate $\tilde{\mathbf{f}}(\boldsymbol{x}_n, \boldsymbol{y}_n, \boldsymbol{\theta}_s) := \left( \log p(\boldsymbol{y}_n|\boldsymbol{x}_n, \boldsymbol{\theta}_s) - \frac{1}{S} \sum_{s'=1}^S \log p(\boldsymbol{y}_n|\boldsymbol{x}_n, \boldsymbol{\theta}_{s'}) \right) \in \mathbb{R}^S$, $\boldsymbol{\theta}_{s'} \sim p(\boldsymbol{\theta}|\boldsymbol{u}, \boldsymbol{z}; \boldsymbol{v})$, (*ii*) computing correlations with the residual error of the total data log-likelihood $\boldsymbol{r}(\boldsymbol{v}) = \mathbf{f}^T(1 - \boldsymbol{v})$, and (*iii*) making a greedy selection of the point that maximizes the correlation between the two: $n^\star \in \underset{n \in [N]}{\arg\max} \begin{cases} |\mathrm{Corr}(\tilde{\mathbf{f}}, \boldsymbol{r})| & v_n > 0 \\ \mathrm{Corr}(\tilde{\mathbf{f}}, \boldsymbol{r}) & v_n = 0 \end{cases}$ , where

$\mathrm{Corr}(\tilde{\mathbf{f}}, \boldsymbol{r}) := \mathrm{diag} \left[ \frac{1}{S} \sum_s \tilde{\mathbf{f}}_s \tilde{\mathbf{f}}_s^T \right]^{-\frac{1}{2}} \left( \frac{1}{S} \sum \tilde{\mathbf{f}}_s \boldsymbol{r}_s^T \right)$. Subsequently, the vector $\boldsymbol{v}$ gets optimized via minimizing the KL divergence between the coreset and the true posterior. For the variational family of coresets, under the assumption that $\boldsymbol{\theta}$ is sampled from the *true* coreset posterior, the formula of the gradient of the objective in Eq. (3) can be derived, and approximated via resampling from the coreset over the course of $\boldsymbol{v}$ optimization. At the end of the optimization, the extracted points and weights are given as input to an approximate inference algorithm—commonly a Laplace approximation—which provides the final estimate of the posterior.

## 3 Black-box coreset Variational Inference

In order to derive a general PSVI objective, we observe that we first need to posit a variational family involving the pseudodata appropriately. Let's assume the (intractable) variational family $q(\boldsymbol{\theta}|\boldsymbol{u}, \boldsymbol{z}) := p(\boldsymbol{\theta}|\boldsymbol{u}, \boldsymbol{z})$, which has the following particular structure: it is parametrized as the posterior over $\boldsymbol{\theta}$ as estimated given inducing points $\boldsymbol{u}, \boldsymbol{z}$. In this case, $q(\boldsymbol{\theta}|\boldsymbol{u}, \boldsymbol{z})$ can be thought of as a *variational program* (see [44]) with variational parameters $\phi = \{\boldsymbol{u}, \boldsymbol{z}\}$ and expressed as:

$$q(\boldsymbol{\theta}|\boldsymbol{u}, \boldsymbol{z}) := p(\boldsymbol{\theta}|\boldsymbol{u}, \boldsymbol{z}) = \frac{p(\boldsymbol{z}|\boldsymbol{u}, \boldsymbol{\theta})p(\boldsymbol{\theta})}{p(\boldsymbol{z}|\boldsymbol{u})}, \tag{4}$$

where the marginal likelihood term $p(\boldsymbol{z}|\boldsymbol{u})$ is typically intractable.

Utilizing this for the formulation of a pseudo-coreset variational inference algorithm leads to the following lower bound on the marginal likelihood of the *true* data:

$$\text{ELBO}_{\text{PSVI}}(\boldsymbol{u}, \boldsymbol{z}) = \mathbb{E}_{q(\boldsymbol{\theta}|\boldsymbol{u},\boldsymbol{z})} \left[ \log \frac{p(\boldsymbol{y}|\boldsymbol{x}, \boldsymbol{\theta})p(\boldsymbol{\theta})}{q(\boldsymbol{\theta}|\boldsymbol{u}, \boldsymbol{z})} \right]. \tag{5}$$

Performing VI involves maximizing Eq. (5) with respect to the parameters $\{\boldsymbol{u}, \boldsymbol{z}\}$. Scrutinizing this objective reveals that the posterior $q(\boldsymbol{\theta}|\boldsymbol{u}, \boldsymbol{z}) := p(\boldsymbol{\theta}|\boldsymbol{u}, \boldsymbol{z})$ is used in two locations: **first**, as the *sampling* distribution for $\boldsymbol{\theta}$, **second** to evaluate log-densities of samples (*scoring*). This objective can only be evaluated if the posterior $p(\boldsymbol{\theta}|\boldsymbol{u}, \boldsymbol{z})$ can be readily evaluated in closed form and sampled from directly. This causes this expression to remain intractable but for the simplest of cases, and reveals the need to make it computable for any sample $\boldsymbol{\theta}$.

### 3.1 Bayesian coresets for intractable posteriors

We identified two key problems to evaluate Eq. (5) involving $q(\boldsymbol{\theta}|\boldsymbol{u}, \boldsymbol{z})$.

**Sampling**  While we cannot sample from the expectation over $q(\boldsymbol{\theta}|\boldsymbol{u}, \boldsymbol{z})$, we can draw $K$ samples from a tractable parametrized distribution $\boldsymbol{\theta}_k \sim r(\boldsymbol{\theta}; \boldsymbol{\psi})$ with variational parameters $\boldsymbol{\psi}$ (i.e. any suitable variational family). However, we cannot plug these samples from $r$ into Eq. (5) directly, as they do not follow the exact distribution called for. To overcome this, we can leverage a self-normalized importance sampling (IS) correction [39] to obtain the desired approximate samples from the coreset posterior. We thus estimate $q(\boldsymbol{\theta}|\boldsymbol{u}, \boldsymbol{z})$ via importance sampling by re-weighting samples $\boldsymbol{\theta}$ from the tractable distribution $r(\boldsymbol{\theta}; \boldsymbol{\psi})$. We denote the resulting implicit distribution $q(\boldsymbol{\theta}|\boldsymbol{u}, \boldsymbol{z}; \boldsymbol{\psi})$.

We assign the samples *unnormalized* weights $w_k$ (noting that $p(\boldsymbol{z}|\boldsymbol{u}, \boldsymbol{\theta}_k) := \prod_i^M p(\boldsymbol{z}_i|\boldsymbol{u}_i, \boldsymbol{\theta}_k)$):

$$w_k = \frac{p(\boldsymbol{z}|\boldsymbol{u}, \boldsymbol{\theta}_k)p(\boldsymbol{\theta}_k)}{r(\boldsymbol{\theta}_k; \boldsymbol{\psi})}, \tag{6}$$

and denote the corresponding *normalized* weights $\tilde{w}_k = w_k / \sum_j w_j$.
Using $q(\boldsymbol{\theta}|\boldsymbol{u}, \boldsymbol{z})$ from Eq. (4), we rewrite Eq. (5) as:

$$\text{ELBO}_{\text{PSVI-IS}}(\boldsymbol{u}, \boldsymbol{z}) = \int_{\boldsymbol{\theta}} q(\boldsymbol{\theta}|\boldsymbol{u}, \boldsymbol{z}) \log \frac{p(\boldsymbol{y}|\boldsymbol{x}, \boldsymbol{\theta})p(\boldsymbol{\theta})}{q(\boldsymbol{\theta}|\boldsymbol{u}, \boldsymbol{z})} d\boldsymbol{\theta} \tag{7}$$

$$= \int_{\boldsymbol{\theta}} \frac{q(\boldsymbol{\theta}|\boldsymbol{u}, \boldsymbol{z})}{r(\boldsymbol{\theta}; \boldsymbol{\psi})} r(\boldsymbol{\theta}; \boldsymbol{\psi}) \log \frac{p(\boldsymbol{y}|\boldsymbol{x}, \boldsymbol{\theta})p(\boldsymbol{\theta})}{q(\boldsymbol{\theta}|\boldsymbol{u}, \boldsymbol{z})} d\boldsymbol{\theta} \approx \sum_{\boldsymbol{\theta}_k \sim r(\boldsymbol{\theta}; \boldsymbol{\psi})} \tilde{w}_k \log \frac{p(\boldsymbol{y}|\boldsymbol{x}, \boldsymbol{\theta}_k)p(\boldsymbol{\theta}_k)}{q(\boldsymbol{\theta}_k|\boldsymbol{u}, \boldsymbol{z})}.$$

**Scoring**  After having tackled sampling, we shift our attention to the second problem, evaluating $q(\boldsymbol{\theta}|\boldsymbol{u}, \boldsymbol{z})$ inside the logarithm. The key problem with scoring $\log q(\boldsymbol{\theta}|\boldsymbol{u}, \boldsymbol{z})$ as defined in Eq. (4) is the intractable marginal probability of the inducing points (IP) $\log p(\boldsymbol{z}|\boldsymbol{u})$. We overcome this by introducing a variational approximation to $\log q(\boldsymbol{\theta}|\boldsymbol{u}, \boldsymbol{z})$ using the same distribution $r(\boldsymbol{\theta}; \boldsymbol{\psi})$ used above and obtain evidence lower bound $\text{ELBO}_{\text{IP}}$ of the IP:

$$\log p(\boldsymbol{z}|\boldsymbol{u}) \geq \text{ELBO}_{\text{IP}}(\boldsymbol{\psi}) = \mathbb{E}_{r(\boldsymbol{\theta}; \boldsymbol{\psi})} \log \frac{q(\boldsymbol{z}|\boldsymbol{u}, \boldsymbol{\theta})p(\boldsymbol{\theta})}{r(\boldsymbol{\theta}; \boldsymbol{\psi})}. \tag{8}$$

**Putting it all together**  We return to Eq. (7) and substitute $q(\boldsymbol{\theta}_k|\boldsymbol{u}, \boldsymbol{z})$, per Eq. (4) to derive the following multi-sample importance sampling lower bound on the evidence of the *observed* data and expose the term $\log p(\boldsymbol{z}|\boldsymbol{u})$:

$$\text{ELBO}_{\text{PSVI-IS}}(\boldsymbol{u}, \boldsymbol{z}) \approx \sum_{\boldsymbol{\theta}_k \sim r} \tilde{w}_k \left[ \log \frac{p(\boldsymbol{y}|\boldsymbol{x}, \boldsymbol{\theta}_k)p(\boldsymbol{z}|\boldsymbol{u})}{p(\boldsymbol{z}|\boldsymbol{u}, \boldsymbol{\theta}_k)} \right] = \sum_{\boldsymbol{\theta}_k \sim r} \tilde{w}_k \log \frac{p(\boldsymbol{y}|\boldsymbol{x}, \boldsymbol{\theta}_k)}{p(\boldsymbol{z}|\boldsymbol{u}, \boldsymbol{\theta}_k)} + \log p(\boldsymbol{z}|\boldsymbol{u}).$$

By lower bounding $\log p(\boldsymbol{z}|\boldsymbol{u})$ with $\text{ELBO}_{\text{IP}}(\boldsymbol{\psi})$, we now propose our final black box objective $\text{ELBO}_{\text{PSVI-IS-BB}}$ as a lower bound to $\text{ELBO}_{\text{PSVI-IS}}(\boldsymbol{u}, \boldsymbol{z})$, under which we can tractably *score* $\boldsymbol{u}, \boldsymbol{z}$:

$$\text{ELBO}_{\text{PSVI-IS}}(\boldsymbol{u}, \boldsymbol{z}) \geq \text{ELBO}_{\text{PSVI-IS-BB}}(\boldsymbol{u}, \boldsymbol{z}, \boldsymbol{\psi})$$

$$= \sum_{\boldsymbol{\theta}_k \sim r} \tilde{w}_k \log \frac{p(\boldsymbol{y}|\boldsymbol{x}, \boldsymbol{\theta}_k)}{p(\boldsymbol{z}|\boldsymbol{u}, \boldsymbol{\theta}_k)} + \mathbb{E}_{\boldsymbol{\theta} \sim r} \left[ \log \frac{p(\boldsymbol{z}|\boldsymbol{u}, \boldsymbol{\theta})p(\boldsymbol{\theta})}{r(\boldsymbol{\theta}; \boldsymbol{\psi})} \right] \tag{9}$$

$$\approx \sum_{\boldsymbol{\theta}_k \sim r} \left[ \tilde{w}_k \log \frac{p(\boldsymbol{y}|\boldsymbol{x}, \boldsymbol{\theta}_k)}{p(\boldsymbol{z}|\boldsymbol{u}, \boldsymbol{\theta}_k)} + \frac{1}{K} \log \frac{p(\boldsymbol{z}|\boldsymbol{u}, \boldsymbol{\theta}_k)p(\boldsymbol{\theta}_k)}{r(\boldsymbol{\theta}_k; \boldsymbol{\psi})} \right].$$

We now remind the reader out that by introducing $\text{ELBO}_{\text{PSVI-IS}}$ we lower bounded $\log p(\boldsymbol{y}|\boldsymbol{x})$, and that $\text{ELBO}_{\text{PSVI-IS-BB}}$ rigorously lower bounds $\text{ELBO}_{\text{PSVI-IS}}$, showing that our proposed fully tractable black box objective $\text{ELBO}_{\text{PSVI-IS-BB}}$ is a rigorous variational objective for approximating $\log p(\boldsymbol{y}|\boldsymbol{x})$:

$$\log p(\boldsymbol{y}|\boldsymbol{x}) \geq \text{ELBO}_{\text{PSVI-IS}}(\boldsymbol{u}, \boldsymbol{z}) \geq \text{ELBO}_{\text{PSVI-IS-BB}}(\boldsymbol{u}, \boldsymbol{z}, \boldsymbol{\psi}).$$

Maximizing $\text{ELBO}_{\text{PSVI-IS-BB}}$ leads to optimizing parameters $\boldsymbol{\psi}$ and thus *adapting* the proposals $r$ for the importance weighting throughout inference, within the confines of the chosen variational family. This is a common approach when blending variational inference and Monte Carlo corrections and can be thought of as specifying an implicit variational family. Depending on the structure of the model one might elect to replace importance sampling with richer Monte Carlo schemes here. We also note that parameters $\boldsymbol{\psi}$ serve maximization of evidence for inducing points given fixed $\{\boldsymbol{u}, \boldsymbol{z}\}$, while parameters $\boldsymbol{\phi} = \{\boldsymbol{u}, \boldsymbol{z}\}$ are adapted to maximize evidence over observed data $\{\boldsymbol{x}, \boldsymbol{y}\}$ which need to account for during optimization of said parameters in Section 3.4. Studying this objective further reveals that replacing our pseudodata dependent importance weights with a uniform distribution $\tilde{w}_k = 1/K, \, k = 1, \ldots, K$, or using a single-sample objective (i.e. in the degenerate case when $K = 1$), allows us to recover the classical ELBO of Eq. (2), cancelling the pseudodata likelihood terms in the variational objective.[2] The setting where we apply this black-box estimator to the basic PSVI algorithm for inferring locations $\boldsymbol{\phi}$ and model variables $\boldsymbol{\psi}$ will be denoted *BB PSVI*.

### 3.2 Variational families: (un)weighted inducing points

When studying the objective in Eq. (5) it quickly becomes evident that the exact amount of the selected coreset datapoints defines the available evidence for that particular posterior construction and as such is a quantity of interest. We will consider this quantity *pseudo-evidence*.

For notational brevity so far we have considered the variational parameters $\boldsymbol{\phi}$ to represent pseudodata directly, while rescaling using the fixed data compression ratio to maintain invariance of the total pseudo-evidence to the coreset size. We will be calling this setting *unweighted pseudodata*.

One choice we can make now is to ensure that the available pseudo-evidence is invariant to the chosen size of the coreset support and potentially equal to the regular dataset. To achieve that, we set:

$$q(\boldsymbol{\theta}|\boldsymbol{u}, \boldsymbol{z}) := p(\boldsymbol{\theta}|\boldsymbol{u}, \boldsymbol{z}) = \frac{p(\boldsymbol{z}|\boldsymbol{u}, \boldsymbol{\theta})^{N/M} p(\boldsymbol{\theta})}{p(\boldsymbol{z}|\boldsymbol{u})} = \frac{\prod^M p(\boldsymbol{z}_i|\boldsymbol{u}_i, \boldsymbol{\theta})^{N/M} p(\boldsymbol{\theta})}{p(\boldsymbol{z}|\boldsymbol{u})}, \tag{10}$$

with $p(\boldsymbol{z}|\boldsymbol{u})$ similarly reweighted via the selected observational *data compression ratio* $N/M$, i.e. for continuous parameter spaces $p(\boldsymbol{z}|\boldsymbol{u}) := \int p(\boldsymbol{z}|\boldsymbol{\theta}, \boldsymbol{u})^{N/M} p(\boldsymbol{\theta}) d\boldsymbol{\theta}$.

We may also consider a scenario where we have more generally *weighted pseudodata*, which involves an additional learnable parameter $\boldsymbol{v}_i \geq 0$ per datapoint $\boldsymbol{u}_i$ and permits the following interpretation: $\boldsymbol{v}_i \sim \text{Multi}(\boldsymbol{v})$. Sampling repeatedly from this distribution yields a collection of inducing points with their frequencies governed by their respective weights:

$$q(\boldsymbol{\theta}|\boldsymbol{v}, \boldsymbol{u}, \boldsymbol{z}) := p(\boldsymbol{\theta}|\boldsymbol{v}, \boldsymbol{u}, \boldsymbol{z}) = \frac{p(\boldsymbol{\theta}) \prod p(\boldsymbol{z}_i|\boldsymbol{u}_i, \boldsymbol{\theta})^{v_i}}{p(\boldsymbol{z}|\boldsymbol{v}, \boldsymbol{u})}. \tag{11}$$

Simplifying the notation for weighted log-likelihoods throughout, we denote:

$$\log p(\boldsymbol{z}_i|v_i, \boldsymbol{u}_i, \boldsymbol{\theta}) := v_i \log p(\boldsymbol{z}_i|\boldsymbol{u}_i, \boldsymbol{\theta}). \tag{12}$$

---

[2]The nested variational program Eq. (8) still allows propagation of non-zero gradients w.r.t. the pseudodata.

We can now also combine the ideas of learning the weights of pseudodata and controlling the amount of pseudoevidence by posing parametrizations where $v_i$ is non-negative and sums to 1 (e.g. $v_i := \texttt{softmax}(\beta_i)$, $i = 1, \ldots, M$), and adding global variational parameters to optimize the magnitude of total evidence over the pseudodata (e.g. $\log p(\boldsymbol{z}|\boldsymbol{v}, \boldsymbol{u}, \boldsymbol{\theta}, \alpha) := \alpha \boldsymbol{v}^T \log p(\boldsymbol{z}|\boldsymbol{u}, \boldsymbol{\theta}))$, where $\alpha$ can be set by hand (i.e. to $N/M$) or be an extra learnable quantity. We will explore various such choices for coreset variational family parametrizations $\boldsymbol{\phi} = \{\boldsymbol{u}, \boldsymbol{z}, \boldsymbol{v}, \alpha\}$ in our experiments and the effects they have on learning.

### 3.3 Incremental and Batch Black-Box Sparse VI

The estimator we derive can also be used to design a black-box version of the Sparse VI algorithm introduced in [10] (see Sec. 2.2), by the insight that it requires only the variational parameters for coreset weights to be updated while using copies of real datapoints as coreset locations. This unlocks various choices to the overall algorithmic flow. The original incremental construction scheme can become black-box via modifying how the approximate posterior on the coreset data gets computed, and introducing a generalized objective for the optimization of model evidence involving the coreset parameters. In this construction (*BB Sparse VI*), we posit a variational family $r(\boldsymbol{\theta}; \boldsymbol{\psi})$ (e.g. mean-field variational distributions) and maximize the corresponding ELBO computed on the weighted datapoints of the coreset. Using the extracted variational approximation, we draw samples from the coreset and correct for the model. Next, over the greedy selection step, we use the samples and importance weights to compute the centered log-likelihood vectors $\tilde{\mathbf{f}}(\boldsymbol{x}_n, \boldsymbol{y}_n, \boldsymbol{\theta}_s) := \log p(\boldsymbol{y}_n|\boldsymbol{x}_n, \boldsymbol{\theta}_s) - \sum_{s'} w_{s'} \log p(\boldsymbol{y}_n|\boldsymbol{x}_n, \boldsymbol{\theta}_{s'})$, and select the next datapoint via the greedy correlation maximization criterion. Subsequently, to refine the weight vector towards the true data posterior, we maximize w.r.t. $\boldsymbol{v}, \boldsymbol{\psi}$ an extension of the PSVI ELBO per Eq. (9), where the coreset data log-likelihood is multiplied by the weight vector $\boldsymbol{v}$ as in Eq. (12):

$$\text{ELBO}_{\text{Sparse-BBVI}}(\boldsymbol{v}, \boldsymbol{\psi}) = \sum_{\boldsymbol{\theta}_k \sim r} \left[ \tilde{w}_k \log \frac{p(\boldsymbol{y}|\boldsymbol{x}, \boldsymbol{\theta}_k)}{p(\boldsymbol{y}|\boldsymbol{v}, \boldsymbol{x}, \boldsymbol{\theta}_k)} + \frac{1}{K} \log \frac{p(\boldsymbol{y}|\boldsymbol{v}, \boldsymbol{x}, \boldsymbol{\theta}_k)p(\boldsymbol{\theta}_k)}{r(\boldsymbol{\theta}_k; \boldsymbol{\psi})} \right]. \quad (13)$$

The importance sampling scheme for the posterior is defined identically to Eq. (6), after substituting the likelihood term with $p(\boldsymbol{y}|\boldsymbol{v}, \boldsymbol{x}, \boldsymbol{\theta}_k)$ for each sample $\boldsymbol{\theta}_k \sim r(\boldsymbol{\theta}_k; \boldsymbol{\psi})$. At each iteration, this variational objective (which, following similar reasoning with the previous section, can be shown to be a lower bound of the evidence), can be maximized wrt the coreset and variational parameters $\boldsymbol{v}, \boldsymbol{\psi}$. Moreover, we can omit the incremental inclusion of points to the coreset, and consider a *batch* version of this construction, where we initialise at a random subset of the original dataset, keep coreset point locations fixed, and optimize only the weights attached to the coreset support using Eq. (13) with non-negativity constraints. Finally, we use the variational approximation fit on the coreset to compute predictive posteriors on unseen datapoints correcting for our importance sampling.

We also define a variant of the algorithm without incremental selection through *pruning* based on the weighting assigned to the coreset points when optimizing the ELBO of Eq. (13). We can start from a large coreset size and, after training, reduce the size of the summary by keeping $K$ samples from a multinomial defined on the coreset points via their so far learned weights, and re-initialising. We provide pseudo-code for the respective methods in Algs. $3-5$ in Supplement B.

### 3.4 Nested optimization

When using a nested optimizer for the maximization of the $\text{ELBO}_{\text{PSVI-IS-BB}}$, rearranging the terms of the objective places our method in the generic cardinality-constrained bilevel optimization framework for data distillation and coreset constructions [6, 26]. VI maximizing the objective of Eq. (5) inside a parametric family $r(\boldsymbol{\theta}; \boldsymbol{\psi})$ can be reformulated as a bilevel optimization problem as:

$$\boldsymbol{v}^\star, \boldsymbol{u}^\star, \boldsymbol{z}^\star = \underset{\boldsymbol{v}, \boldsymbol{u}, \boldsymbol{z}}{\arg\min}\, L(\boldsymbol{v}, \boldsymbol{u}, \boldsymbol{z}, \boldsymbol{\psi}^\star(\boldsymbol{v}, \boldsymbol{u}, \boldsymbol{z}); \boldsymbol{x}, \boldsymbol{y}) \quad \text{s.t.} \quad \boldsymbol{\psi}^\star = \underset{\boldsymbol{\psi}}{\arg\min}\, \ell(\boldsymbol{\psi}, \boldsymbol{v}, \boldsymbol{u}, \boldsymbol{z}), \quad (14)$$

where

$$L(\boldsymbol{v}, \boldsymbol{u}, \boldsymbol{z}, \boldsymbol{\psi}(\boldsymbol{v}, \boldsymbol{u}, \boldsymbol{z})) = -\text{ELBO}_{\text{PSVI-IS}}(\boldsymbol{v}, \boldsymbol{u}, \boldsymbol{z}, \boldsymbol{\psi}) \quad (15)$$

and

$$\ell(\boldsymbol{\psi}, \boldsymbol{v}, \boldsymbol{u}, \boldsymbol{z}) = -\text{ELBO}(\boldsymbol{\psi}, \boldsymbol{v}, \boldsymbol{u}, \boldsymbol{z}) := -\mathbb{E}_{\boldsymbol{\theta}_k \sim r} \log \frac{p(v_i, \boldsymbol{z}|\boldsymbol{u}, \boldsymbol{\theta}_k)p(\boldsymbol{\theta}_k)}{r(\boldsymbol{\theta}_k; \boldsymbol{\psi})}. \quad (16)$$

We optimize via iterative differentiation [29], i.e. trace the gradient computation over the optimization of the variational parameters $\psi$ when solving the inner problem, and use this information at the computation of the outer objective gradient wrt the variational coreset parameters $\phi = \{v, u, z, \alpha\}$.

# 4 Experiments

In this section we evaluate the performance of our inference framework in intractable models and compare against standard variational inference methods and earlier Bayesian coreset constructions, as well as black-box extensions of existing variational coresets that rely on our generalized ELBO Eq. (9).

As a running baseline with unrestricted access to the training data, we use the standard *mean-field* VI with a diagonal covariance matrix. To capture the implications of enforcing data cardinality constraints, we also construct a *random coreset* using a randomly selected data subset with fixed data locations and fixed likelihood multiplicities correcting for the dataset size reduction, which we then use to optimize a mean-field approximate posterior. As earlier work relied on Laplace approximations on coreset data as a black-box approach for sampling for intractable coreset posteriors, we also experiment with Laplace approximations on random data subsets as an extra baseline. We make code available at `www.github.com/facebookresearch/Blackbox-Coresets-VI`.

## 4.1 Logistic regression

First, we perform inference on logistic regression fitting 3 publicly available binary classification datasets [17, 53] with sizes ranging between $10k$ and $150k$ datapoints, and 10 and 128 dimensions. We posit normal priors $\theta \sim \mathcal{N}(0, I)$ and consider mean-field variational approximations with diagonal covariance. In the following, all presented metrics are averaged across 3 independent trials, and we show the means along with the corresponding standard errors.

**Impact of variational parameters** We present the predictive metrics of accuracy and log-likelihood on the test set in Tables 1 and 2. We observe that coreset methods relying on optimizing pseudodata have the capacity to better approximate inference on the full dataset for small coreset sizes regardless of data dimension, reaching the performance of unrestricted mean-field VI with the use of less than 50 weighted points. In contrast, the approximations that rely on existing datapoints are limited by data dimensionality and need to acquire a larger support to reach the performance of full-data VI (with effects being more evident in `webspam`, which has a data dimensionality of 128). Including variational parameterisations for optimal scaling of the coreset likelihood aids inference, offering faster convergence. Omitting to correct for dataset reduction ($v = 1$) has detrimental effects on predictive performance. As opposed to earlier constructions limited by heuristics on coreset posterior approximation throughout optimization, our variational framework enables approximating the full data posterior within the assumed variational family. Evaluation of predictive metrics and computation time requirements across a wider range of coreset sizes is included in the supplement.

**Impact of importance weighting** In Fig. 2 we plot the normalized effective sample size (ESS) of our importance weighting scheme for 10 Monte Carlo samples computed on test data. We observe that the posteriors can benefit from IW, achieving non-trivial ESSs, which tend to converge to close values for large coreset sizes. This finding applies at the testing phase also on ablations of the PSVI variational training that replace the IW scheme with uniform sampling.

**Importance sampling in high dimensions.** Importance sampling can suffer from large variance in high dimensional spaces [39]. Hence, one may be skeptical about the approximation quality the scheme can provide. To this end, in Supplement G we conduct a complementary experiment with a synthetic logistic regression problem. As dimensionality increases, we find that while ESS goes down, it remains non-trivial. More involved sampling methods such as [58] are a promising avenue for leveraging the modular nature of BB PSVI to achieve further performance improvements.

## 4.2 Bayesian Neural Networks

In this section we present inference results on Bayesian neural networks (BNNs), a model class that previous work on Bayesian coresets did not consider due to the absence of a black-box variational estimator. In the first part we perform inference via black-box PSVI on 2-dimensional synthetic data

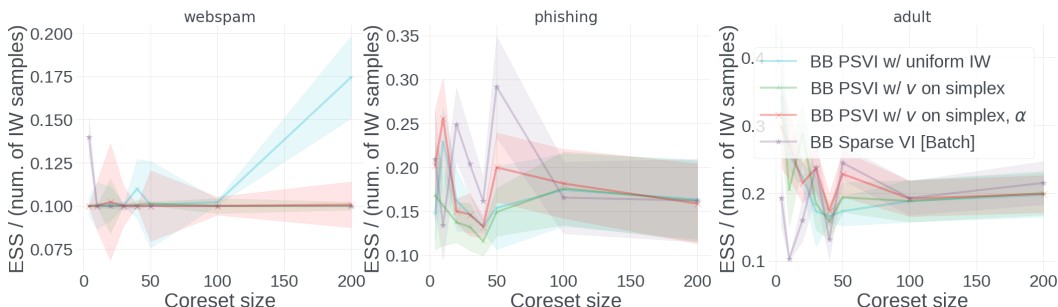

Figure 2: Normalized effective sample size for predictions on test data using 10 Monte Carlo samples from the coreset posterior over increasing coreset size.

Table 1: Test accuracy on 3 logistic regression datasets. (⋆) denotes use of softmax parameterisation for the weights $\boldsymbol{v} = N\texttt{softmax}(\beta)$.

| | | webspam | | | phishing | | | adult | | |
|---|---|---|---|---|---|---|---|---|---|---|
| | $M$ | 10 | 40 | 100 | 10 | 40 | 100 | 10 | 40 | 100 |
| BB PSVI | $\boldsymbol{v}(\star), \boldsymbol{u}$ | **92.2**±0.1 | **92.4**±0.1 | **92.3**±0.0 | 90.1±0.5 | **91.2**±1.0 | 90.4±0.0 | 82.2±0.9 | **83.5**±0.2 | 83.5±0.0 |
| | $\boldsymbol{v}(\star), \boldsymbol{u}, \alpha$ | **92.2**±0.2 | 92.2±0.1 | **92.3**±0.1 | 90.7±0.5 | 90.7±0.9 | 90.4±0.0 | **82.6**±0.9 | **83.5**±0.0 | **83.6**±0.0 |
| | $\boldsymbol{v}, \boldsymbol{u}$ | **92.2**±0.1 | 92.3±0.1 | 92.2±0.1 | 90.3±0.8 | 89.8±0.2 | 90.3±0.1 | 80.3±0.8 | **83.5**±0.0 | 83.5±0.0 |
| | $\boldsymbol{v}{=}\frac{N}{M}\mathbf{1}, \boldsymbol{u}$ | 90.9±0.3 | 91.9±0.1 | 92.2±0.5 | 90.2±0.3 | 88.9±0.7 | **90.5**±0.0 | 81.0±0.4 | **83.5**±0.1 | 83.5±0.5 |
| | unif. IW, $\boldsymbol{v}(\star), \boldsymbol{u}$ | 92.1±0.1 | 92.3±0.1 | 92.2±0.1 | **90.8**±1.0 | 90.6±0.9 | 90.4±0.0 | 82.1±0.8 | **83.5**±0.1 | 83.5±0.0 |
| | w/o IW, $\boldsymbol{v}(\star), \boldsymbol{u}$ | 90.1±0.6 | 92.0±0.1 | **92.3**±0.1 | 90.1±1.4 | 89.2±1.2 | 90.4±0.2 | 77.8±1.5 | **83.5**±0.0 | 83.5±0.1 |
| | $\boldsymbol{v}{=}\mathbf{1}, \boldsymbol{u}$ | 61.1±1.8 | 61.2±4.7 | 61.2±2.2 | 85.8±10.4 | 86.3±11.1 | 87.3±11.5 | 75.0±3.7 | 75.1±1.6 | 75.8±1.2 |
| BB Sparse VI | Batch | 80.1±0.9 | 86.9±0.2 | 88.9±0.4 | 87.6±0.8 | 90.7±0.3 | 90.2±1.2 | 78.1±1.1 | 82.6±0.4 | 83.5±0.7 |
| | Incremental | 60.7±0.0 | 76.9±3.3 | 83.4±4.5 | 88.8±2.1 | 88.9±3.7 | 88.1±2.2 | 78.6±0.1 | 81.9±0.9 | 82.4±0.7 |
| PSVI [31] | | 83.9±0.7 | 88.8±0.4 | 90.3±0.3 | 88.3±1.7 | 89.0±0.9 | 90.1±0.1 | 82.5±0.4 | **83.5**±0.0 | 83.5±0.0 |
| Sparse VI [10] | | 72.8±3.5 | 74.2±4.7 | 74.6±4.8 | 87.0±1.6 | 89.3±0.5 | 90.0±1.4 | 78.1±2.3 | 79.0±3.1 | 80.7±1.9 |
| Subset Laplace | | 63.0±6.4 | 78.8±2.3 | 82.7±1.3 | 72.7±4.1 | 84.7±2.1 | 88.0±2.1 | 64.4±2.2 | 77.0±2.2 | 81.2±1.2 |
| Rand. Coreset | | 71.3±2.9 | 83.6±1.2 | 86.3±0.6 | 81.4±2.7 | 86.1±3.8 | 84.4±1.7 | 75.9±1.1 | 76.2±0.3 | 79.5±0.5 |
| Full MFVI | | | | **92.7**±0 | | | 90.4±0.1 | | | **83.5**±0 |

using single-hidden layer BNNs, while in the latter part we evaluate the performance of our methods on compressing the MNIST dataset using the LeNet architecture [25].

**Simulated datasets** We generate two synthetic datasets with size $1k$ datapoints, corresponding to noisy samples from a half-moon shaped 2-class dataset, and a mixture of 4 unimodal clusters of data each belonging to a different class [24], and use a layer with 20 and 50 units respectively. To evaluate the representation ability of the pseudocoresets we consider two initialization schemes: we initialise the pseudo locations on a random subset equally split across categories, and a random initialization using a Gaussian centered on the means of the empirical distributions. In Fig. 3 we visualize the inferred predictive posterior entropy, along with coreset points locations and relative weights. Regardless of the initialization, BB PSVI has the capacity to optimize the locations of the coreset support according to the statistics of the true data distribution, yielding correct decision boundaries. Importantly, the coreset support consists itself of separable summarizing pseudo points, while the likelihood reweighting variational parameters enable emphasizing on critical regions of the

Table 2: Test negative log. likelihood on 3 logistic regression datasets.

| | | webspam | | | phishing | | | adult | | |
|---|---|---|---|---|---|---|---|---|---|---|
| | $M$ | 10 | 40 | 100 | 10 | 40 | 100 | 10 | 40 | 100 |
| BB PSVI | $\boldsymbol{v}(\star), \boldsymbol{u}$ | 21.7±0.3 | 21.9±0.2 | 21.2±0.2 | 36.1±0.8 | 34.8±3.8 | 26.0±0.1 | **36.1**±0.8 | 34.8±3.8 | **26.0**±0.1 |
| | $\boldsymbol{v}(\star), \boldsymbol{u}, \alpha$ | 22.1±0.5 | 21.6±0.2 | 21.0±0.2 | 33.6±0.6 | 34.2±3.1 | **25.9**±0.1 | 50.2±3.2 | 34.0±0.0 | 33.9±0.0 |
| | $\boldsymbol{v}, \boldsymbol{u}$ | 21.9±0.4 | 21.9±0.1 | 21.6±0.3 | 40.3±2.1 | 48.9±6.2 | 26.0±0.1 | 78.0±6.2 | 34.0±0.1 | 34.0±0.1 |
| | $\boldsymbol{v}{=}\frac{N}{M}\mathbf{1}, \boldsymbol{u}$ | 26.4±1.1 | 23.6±0.3 | 22.3±1.3 | 40.8±0.8 | 57.7±10.9 | **25.9**±0.1 | 79.5±14.5 | **33.9**±0.1 | 33.9±1.4 |
| | unif. IW, $\boldsymbol{v}(\star), \boldsymbol{u}$ | **21.2**±0.2 | **20.8**±0.1 | **20.8**±0.1 | **31.8**±1.8 | **31.3**±2.7 | **25.9**±0.1 | 48.1±2.2 | **33.9**±0.1 | 33.9±0.1 |
| | w/o IW, $\boldsymbol{v}(\star), \boldsymbol{u}$ | 29.8±2.1 | 21.8±0.4 | 21.2±0.3 | 43.8±5.5 | 35.8±6.3 | 26.9±0.5 | 74.9±10.2 | 34.2±0.1 | 34.0±0.3 |
| | $\boldsymbol{v}{=}\mathbf{1}, \boldsymbol{u}$ | 69.4±3.2 | 78.4±4.9 | 70.4±4.7 | 37.5±24.4 | 34.7±23.6 | 38.3±24.8 | 57.7±7.1 | 53.1±3.6 | 49.4±3.0 |
| BB Sparse VI | Batch | 65.0±8.6 | 39.1±1.6 | 35.9±1.2 | 40.2±2.9 | 35.2±0.9 | 25.9±2.1 | 70.8±4.4 | 48.3±1.6 | 33.9±10.2 |
| | Incremental | 86.1±11.0 | 54.0±3.2 | 40.7±2.3 | 33.6±3.6 | 34.8±3.1 | 29.4±2.9 | 45.7±2.2 | 37.6±1.5 | 36.5±0.6 |
| PSVI [31] | | 81.9±8.4 | 42.4±0.6 | 37.2±0.9 | 32.5±3.1 | 51.2±16.9 | 26.0±0.1 | 40.7±2.4 | 34.0±0.1 | 34.0±0.0 |
| Sparse VI [10] | | 53.9±8.0 | 52.6±9.3 | 52.7±9.2 | 44.0±4.0 | 38.3±1.9 | 37.7±2.0 | 45.3±4.9 | 44.1±6.1 | 41.9±4.5 |
| Subset Laplace | | 96.5±19.0 | 46.0±3.9 | 37.8±2.1 | 55.1±4.7 | 43.3±4.4 | 40.9±1.7 | 119.2±16.3 | 55.0±9.2 | 39.4±1.1 |
| Rand. Coreset | | 175.1±25.3 | 76.9±15.5 | 83.1±5.8 | 122.0±13.4 | 156.0±53.0 | 47.6±3.4 | 251.4±32.1 | 178.5±39.5 | 51.8±6.7 |
| Full MFVI | | | | **19.8**±0.0 | | | **26.1**±0.1 | | | **33.9**±0.0 |

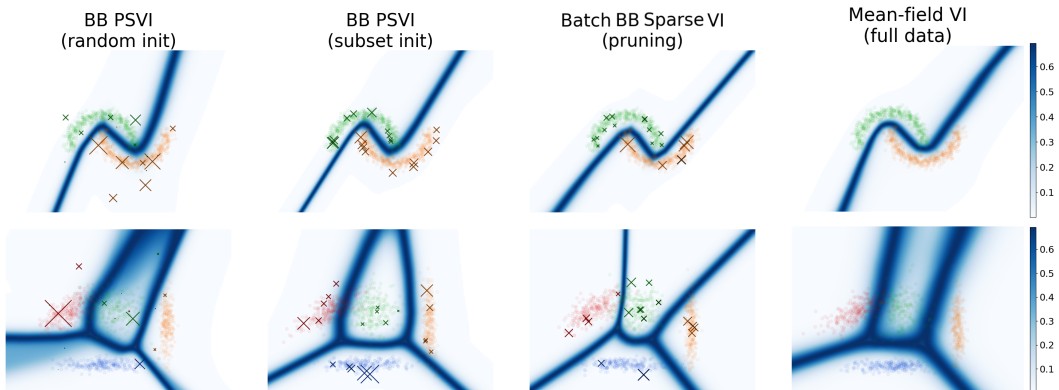

Figure 3: Predictive posterior entropy on simulated datasets. Coreset points are visualised in crosses, with size proportional to their weights.

Table 3: Test accuracy on the MNIST dataset using NNs with the LeNet architecture for our black-box coreset constructions and other learning baselines. Full MFVI accuracy: $99.13 \pm 0.07$

|  | $M$ | 10 | 30 | 50 | 80 | 100 | 250 |
|---|---|---|---|---|---|---|---|
| | $v(\star), \boldsymbol{u}, \boldsymbol{z}, \alpha$ | $\mathbf{53.65} \pm 2.48$ | $\mathbf{76.78} \pm 2.1$ | $\mathbf{85.65} \pm 0.0$ | $89.43 \pm 0.56$ | $\mathbf{91.28} \pm 0.74$ | $\mathbf{93.85} \pm 0.0$ |
| BB PSVI | $v(\star), \boldsymbol{u}, \boldsymbol{z}$ | $52.97 \pm 3.32$ | $76.07 \pm 2.06$ | $85.05 \pm 0.3$ | $\mathbf{89.87} \pm 0.95$ | $90.77 \pm 0.92$ | $93.51 \pm 0.14$ |
| | $v(\star), \boldsymbol{u}, \alpha$ | $40.43 \pm 0.66$ | $63.15 \pm 0.14$ | $77.54 \pm 1.32$ | $84.52 \pm 0.18$ | $85.47 \pm 0.21$ | $90.98 \pm 0.33$ |
| | $v(\star), \boldsymbol{u}$ | $40.69 \pm 0.84$ | $62.8 \pm 0.02$ | $76.85 \pm 0.86$ | $84.52 \pm 0.31$ | $86.03 \pm 0.6$ | $91.3 \pm 0.37$ |
| BB Sparse VI [Batch] | | $43.7 \pm 2.14$ | $68.41 \pm 0.6$ | $77.21 \pm 2.43$ | $83.88 \pm 0.44$ | $85.65 \pm 0.17$ | $91.54 \pm 0.16$ |
| Dataset Condens. [57] | | - | - | - | - | $93.9 \pm 0.6$ | - |
| SLDD [49] | | - | - | - | - | $82.7 \pm 2.8$ | - |
| Dataset Distill. [54] | | - | - | - | - | $79.5 \pm 8.1$ | - |
| Random Coreset | | $42.17 \pm 1.74$ | $68.67 \pm 3.21$ | $78.75 \pm 2.17$ | $86.24 \pm 0.25$ | $87.44 \pm 0.32$ | $92.1 \pm 0.65$ |

landscape, assigning larger weights to pseudopoints lying close to the boundaries among the classes. Moreover, our pruning scheme which makes use of batch BB Sparse VI is able to gradually compress a large coreset of 250 datapoints to a compact performant subset of 20 datapoints arranged in critical locations for learning the Bayesian posterior.

**MNIST**  In this part we assess the approximation quality of large-scale dataset compression for BNNs via coresets. We compare the predictive performance of black-box PSVI against standard mean-field VI, random coresets and frequentist methods relying on learnable synthetic data, namely dataset distillation w/ and w/o learnable soft labels [49, 54], and data condensation [57]. We can see that BB PSVI consistently outperforms the random coreset baseline for coreset sizes in the orders of few tens or hundreds, however the gap narrows towards 250 coreset points (Table 3). Moreover, at size 100 our construction provides better test accuracy compared to two frequentist approaches, enjoying the additional advantage of providing uncertainty estimates.

**Continual learning with Bayesian coresets**  As discussed in the introduction, we envision coresets being leveraged as a tool beyond scaling sampling methods to large datasets. One such example setting is continual learning, where past methods have encoded statistical information in the form of approximate posterior distributions over the weights [36, 47]. While [36] leverages (randomly selected) coresets, these necessarily lose information about the data due to the approximate nature of their closed-form posteriors. Bayesian coresets, in contrast do not make use of an explicit closed-form posterior, as they rely on the implicit posterior induced by the coreset, representing the prior information by mixing the pseudo data at frequency equal to their weight with the new data. We showcase such a continual learning method on the four-class synthetic problem from above in Fig. 10 with the difference that each class arrives sequentially after the initial two and prior classes are not revisited except for the coresets (details in Supplement D). Non-Bayesian approaches to continual learning with coresets have been investigated in more depth in [4, 6, 55].

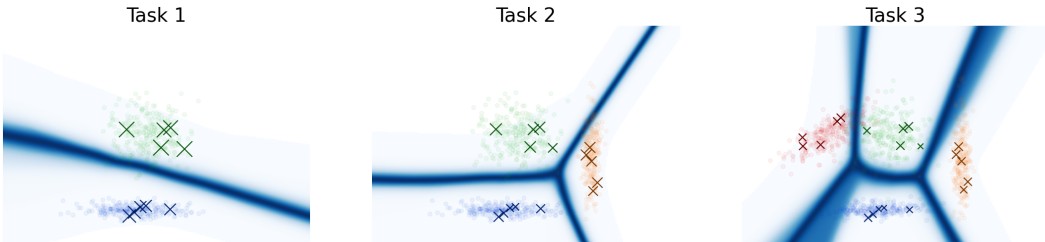

Figure 4: Incremental learning: a coreset is constructed by incrementally fitting a BNN to 3 classification tasks, starting with 10 coreset points and subsequently increasing to 15 and 20.

# 5    Related work and discussion

**Coresets and data pruning beyond Bayesian inference**    The paradigm of coresets emerged from seminal work in computational geometry [1, 18], and has since found broad use in data-intensive machine learning, including least mean square solvers [28], dimensionality reduction [20], clustering [2, 19, 27] and maximum likelihood estimation [3, 32]. Coreset ideas have recently found applications in active learning [42, 7], continual learning [4, 6] and robustness [30, 33]. Recent works in neural networks have investigated heuristics for pruning datasets, including the misclassification events of datapoints over learning [52], or gradient scores in early stages of training [41].

**Pseudodata for learning**    Despite the breakthroughs of classical coreset methods, the common confine of selecting the coresets among the constituent datapoints might lead to large summary sizes in practice. Exempt from this limitation are methods that synthesize informative pseudo-examples for training that can summarize statistics lying away from the data manifold. These appear in the literature under various terms, including dataset distillation [26, 29, 54], condensation [57], dataset meta-learning [37] and inducing points [38]. Moreover, recent work [51] has demonstrated that incorporating learnable pseudo-inputs as hyperparameters of the model prior can enhance inference in variational auto-encoders. Learnable pseudodata that resemble the sufficient statistics of the original dataset have also been used in herding [13] and scalable GP inference methods [48, 50].

**Variational inference, importance weighting, and Monte Carlo objectives**    Importance weighting can provide tighter lower bounds of the data log-likelihood for training variational auto-encoders [9], resulting in richer latent representations [14]. In later work, [15] propose an importance weighting VI scheme for general purpose probabilistic inference, while [44, 45] introduce variational programs as programmatic ways to specify a variational distribution. Higher dimensional proposals are commonly handled via Sequential Monte Carlo techniques [16, 34], while [58] develops a family of methods for learning proposals for importance samplers in nested variational approximations. **Two forces** are at play when considering our approximate inference scheme: the variational family $r$ and the sampling scheme used to estimate $q(\boldsymbol{\theta}|\boldsymbol{u}, \boldsymbol{z}; \boldsymbol{\psi})$. These two interact nontrivially, since $r$ forms the proposal for the importance sampling scheme, and this inference yields gradients for the formation of the coreset. For hard inference problems richer sampling strategies (e.g. SMC) and variational families may yield useful coreset and posterior beliefs over the model faster and with more accuracy.

# 6    Conclusion

We have developed a framework for novel black-box constructions of variational coresets that can be applied at scale to large datasets as well as richly structured models, and result in effective model-conditional dataset summarization. We overcome the intractability of previous inference algorithms with our proposed black-box objectives, which we develop for both Sparse VI as well as PSVI, and propose a suite of algorithms utilizing them.

Through experiments on intractable models, we show that the inferred coresets form a useful summary of a dataset with respect to the learning problem at hand, and are capable to efficiently compress information for Bayesian posterior approximations. In future work, we plan to extend this inference toolbox for privacy purposes and non-traditional learning settings, e.g. continual learning, explore alternative objectives, and richer models of coresets.

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
