# A  Probabilistic models for classification with learnable soft labels

Including the labels of the pseudodata to the coreset parameters allows more efficient compression, as our summarizing data now live in an expanded space spanning uncertain pseudolabels. In this case we have to adapt our variational objective to capture the divergence between the distribution over the labels corresponding to the soft labeling of the coreset and the predictive distribution under the current variational posterior. This term can be upper bounded per coreset point indexed by $m$ by interpreting the soft labels as categorical probabilities and minimising the KL divergence between the corresponding distribution and the predictive distribution under the approximate posterior as:

$$
\begin{aligned}
\mathrm{D_{KL}}\left(p(y_m|\boldsymbol{z}_m)||\mathbb{E}_{q(\boldsymbol{\theta})}\left[p(y_m|\boldsymbol{u}_m,\boldsymbol{\theta})\right]\right) &= \mathbb{E}_{p(y_m|\boldsymbol{z}_m)}\left[\log p(y_m|\boldsymbol{z}_m) - \log\mathbb{E}_{q(\boldsymbol{\theta})}\left[p(y_m|\boldsymbol{u}_m,\boldsymbol{\theta})\right]\right] \\
&\leq \mathbb{E}_{p(y_m|\boldsymbol{z}_m)}\left[\log p(y_m|\boldsymbol{z}_m) - \mathbb{E}_{q(\boldsymbol{\theta})}\left[\log p(y_m|\boldsymbol{u}_m,\boldsymbol{\theta})\right]\right] \\
&= \mathbb{E}_{q(\boldsymbol{\theta})}\left[\mathbb{E}_{p(y_m|\boldsymbol{z}_m)}\left[\log p(y_m|\boldsymbol{z}_m) - \log p(y_m|\boldsymbol{u}_m,\boldsymbol{\theta})\right]\right] \\
&= \mathbb{E}_{q(\boldsymbol{\theta})}\left[\mathrm{D_{KL}}\left(p(y_m|\boldsymbol{z}_m)||p(y_m|\boldsymbol{u}_m,\boldsymbol{\theta})\right)\right], \quad (17)
\end{aligned}
$$

where the inequality follows from Jensen's inequality and $-\log$ being convex. In this context $y_m$ is seen as a random variable and not a fixed label value. If the $\boldsymbol{z}_m$ probabilities are all 1s, corresponding to the fixed label case, this expression reduces to the expected negative log-likelihood of those labels under the approximate posterior and we recover the negative log likelihood part of the classical (negative) ELBO.

# B  Pseudocode for Sparse VI vs Black-box Sparse VI

For clarity we include an algorithmic description of the incremental version of our black-box scheme for Sparse VI (Algorithm 3), to be contrasted with the earlier construction of [10] which relies on estimates of the analytical gradient of the KL divergence between the coreset and the true posterior (Algorithm 1). The batch version of the black-box Sparse VI construction is presented in Algorithm 4, where the entire support of the coreset gets jointly optimized without a greedy selection step. The corresponding pruning strategy, which is designed to shrink a given coreset size to a coreset with larger sparsity, is described in Algorithm 5. We typically opt for parameterisations of $\boldsymbol{v}$ that enforce non-negativity (e.g. via the `softmax` function), hence projection over gradient updates is generally not required.

**Algorithm 1** Sparse VI
***

$\boldsymbol{v} \leftarrow \boldsymbol{0} \in \mathbb{R}^M, \ \mathcal{I} \leftarrow \emptyset \quad \triangleright$ Initialise to the empty coreset

**for** $k = 1, \dots, K$ **do**

$\quad (\boldsymbol{\theta})_{s=1}^S \overset{\text{i.i.d.}}{\sim} \pi_{\boldsymbol{v}} \quad \triangleright$ Take $S$ samples from current coreset posterior

$\quad \mathcal{B} \sim \mathsf{UnifSubset}\,([N], B) \quad \triangleright$ Obtain a minibatch of $B$ datapoints from the full dataset

$\quad \triangleright$ Compute likelihood vectors over the coreset and minibatch datapoints for each sample

$\quad g_s \leftarrow \left( f(x_m, \boldsymbol{\theta}_s) - \frac{1}{S} \sum_{r=1}^S f(x_m, \boldsymbol{\theta}_r) \right)_{m \in \mathcal{I}} \in \mathbb{R}^M$

$\quad g'_s \leftarrow \left( f(x_b, \boldsymbol{\theta}_s) - \frac{1}{S} \sum_{r=1}^S f(x_b, \boldsymbol{\theta}_r) \right)_{b \in \mathcal{B}} \in \mathbb{R}^B$

$\quad \triangleright$ Get empirical estimates of correlation over the coreset and minibatch datapoints

$\quad \widehat{\mathrm{Corr}} \leftarrow \mathrm{diag} \left[ \frac{1}{S} \sum_{s=1}^S g_s g_s^T \right]^{-\frac{1}{2}} \left( \frac{1}{S} \sum_{s=1}^S g_s \left( \frac{N}{B} 1^T g'_s - \boldsymbol{v}^T g_s \right) \right) \in \mathbb{R}^M$

$\quad \widehat{\mathrm{Corr}}' \leftarrow \mathrm{diag} \left[ \frac{1}{S} \sum_{s=1}^S g'_s g_s'^T \right]^{-\frac{1}{2}} \left( \frac{1}{S} \sum_{s=1}^S g'_s \left( \frac{N}{B} 1^T g'_s - \boldsymbol{v}^T g_s \right) \right) \in \mathbb{R}^B$

$\quad \triangleright$ Select next point to be attached via max. correlation with the residual error vector

$\quad n^\star \leftarrow \arg\max_{n \in [m] \cup [B]} \left( \left| \widehat{\mathrm{Corr}} \right| \cdot \mathbb{1}[n \in \mathcal{I}] + \widehat{\mathrm{Corr}}' \cdot \mathbb{1}[n \notin \mathcal{I}] \right), \ \ \mathcal{I} \leftarrow \mathcal{I} \cup \{n^\star\}$

$\quad \triangleright$ Optimize the weights $\boldsymbol{v}$ via proj. gradient descent using estimates of the analytical gradient

$\quad$ **for** $t = 1, \dots, T$ **do**

$\quad\quad (\boldsymbol{\theta})_{s=1}^S \overset{\text{i.i.d.}}{\sim} \pi_{\boldsymbol{v}}$

$\quad\quad \triangleright$ Compute likelihood vectors over the coreset and minibatch datapoints for each sample

$\quad\quad g_s \leftarrow \left( f(x_m, \boldsymbol{\theta}_s) - \frac{1}{S} \sum_{r=1}^S f(x_m, \boldsymbol{\theta}_r) \right)_{m \in \mathcal{I}} \in \mathbb{R}^M$

$\quad\quad g'_s \leftarrow \left( f(x_b, \boldsymbol{\theta}_s) - \frac{1}{S} \sum_{r=1}^S f(x_b, \boldsymbol{\theta}_r) \right)_{b \in \mathcal{B}} \in \mathbb{R}^B$

$\quad\quad \hat{\nabla}_{\boldsymbol{v}} \leftarrow -\frac{1}{S} \sum_{s=1}^S g_s \left( \frac{N}{B} 1^T g'_s - \boldsymbol{v}^T g_s \right) \quad \triangleright$ Compute MC gradients for variational parameters

$\quad\quad \boldsymbol{v} \leftarrow \max(\boldsymbol{v} - \gamma_t \hat{\nabla}_{\boldsymbol{v}}, 0) \quad \triangleright$ Take a projected stochastic gradient step

$\quad$ **end for**

$\quad$ **return** $\boldsymbol{v}$

**end for**

At test time use $(\boldsymbol{\theta})_{s=1}^S \overset{\text{i.i.d.}}{\sim} \pi_{\boldsymbol{v}}$ to compute the predictive posterior
***

**Algorithm 2** Correction for sampling under coreset
***

$\triangleright$ Forward inducing and true data through the model using $S$ samples from the coreset variational approximation $\boldsymbol{\theta} \sim r(\boldsymbol{\theta}; \boldsymbol{\psi})$ and compute $p(\boldsymbol{y}|\boldsymbol{x}, \boldsymbol{\theta}) \in \mathbb{R}^{N \times S}, p(\boldsymbol{z}|\boldsymbol{v}, \boldsymbol{u}, \boldsymbol{\theta}) \in \mathbb{R}^{M \times S}$

$\triangleright$ Compute importance weights for the coreset samples

$$w_s = \sum_i^M [\log p(\boldsymbol{z}_i | \boldsymbol{u}_i, \boldsymbol{v}_i, \boldsymbol{\theta}_s)] - \mathrm{D}_{\mathrm{KL}} \left( r(\boldsymbol{\theta}_s; \boldsymbol{\psi}) || p(\boldsymbol{\theta}_s) \right), \ \tilde{w}_s = \frac{w_s}{\sum_{s'} w_{s'}} \ \text{ for } s = 1, \dots, S.$$

**return** Importance weights $\tilde{\boldsymbol{w}}$ and correct marginals using

$$\hat{p}(\boldsymbol{y}|\boldsymbol{x}) = p(\boldsymbol{y}|\boldsymbol{x}, \boldsymbol{\theta}) \tilde{\boldsymbol{w}}$$
***

**Algorithm 3** Black-Box Sparse VI [Incremental] (Ours)

---

$\boldsymbol{v} \leftarrow \mathbf{0} \in \mathbb{R}^M, \quad g \leftarrow \mathbf{0} \in \mathbb{R}^{S \times M}, \quad g' \leftarrow \mathbf{0} \in \mathbb{R}^{S \times B}, \quad \mathcal{I} \leftarrow \emptyset \quad \triangleright$ Initialise to the empty coreset and pick initial $\boldsymbol{\psi}$ for $q$

**for** $k = 1, \ldots, K$ **do**

   $\mathcal{B} \sim \mathsf{UnifSubset}\left([N], B\right) \quad \triangleright$ Obtain a minibatch of $B$ datapoints from the full dataset

   $\triangleright$ Optimize wrt $\boldsymbol{\psi}$ on active coreset weighted datapoints with fixed data locations using the classical ELBO

   $(\boldsymbol{\theta})_{s=1}^S \sim r(\boldsymbol{\theta}; \boldsymbol{\psi}), \quad g_s \in \mathbb{R}^M, \quad g'_s \in \mathbb{R}^B \quad \triangleright$ Forward each datapoint through the statistical model correcting via importance weighting $\tilde{\mathbf{w}}$ to obtain the centered likelihood vectors using a batch $S$ of samples from $r$ (Algorithm 2)

   $\triangleright$ Get empirical estimates of correlation over the coreset and minibatch datapoints

   $\widehat{\mathrm{Corr}} \leftarrow \mathrm{diag}\left[\frac{1}{S}\sum_{s=1}^S g_s g_s^T\right]^{-\frac{1}{2}} \left(\frac{1}{S}\sum_{s=1}^S g_s \left(\frac{N}{B}\mathbf{1}^T g'_s - \boldsymbol{v}^T g_s\right)\right) \in \mathbb{R}^M$

   $\widehat{\mathrm{Corr}}' \leftarrow \mathrm{diag}\left[\frac{1}{S}\sum_{s=1}^S g'_s g_s'^T\right]^{-\frac{1}{2}} \left(\frac{1}{S}\sum_{s=1}^S g'_s \left(\frac{N}{B}\mathbf{1}^T g'_s - \boldsymbol{v}^T g_s\right)\right) \in \mathbb{R}^B$

   $\triangleright$ Select next point to be attached via max. correlation with the residual error vector

   $n^\star \leftarrow \arg\max_{n \in [m] \cup [B]} \left(\left|\widehat{\mathrm{Corr}}\right| \cdot \mathbb{1}[n \in \mathcal{I}] + \widehat{\mathrm{Corr}}' \cdot \mathbb{1}[n \notin \mathcal{I}]\right), \quad \mathcal{I} \leftarrow \mathcal{I} \cup \{n^\star\}$

   $\triangleright$ Optimize the weights vector $\boldsymbol{v}$ via projected gradient descent

   **for** $t = 1, \ldots, T$ **do**

      $\mathcal{B} \sim \mathsf{UnifSubset}\left([N], B\right)$

      $(\boldsymbol{\theta})_{i=1}^S \sim r(\boldsymbol{\theta}; \boldsymbol{\psi}) \quad \triangleright$ Resample and compute importance weights using Algorithm 2

      $\triangleright$ Compute the outer gradient wrt $\boldsymbol{v}$ using the gradient information of the inner optimization wrt $\boldsymbol{\psi}$

      $\hat{\nabla}_{\boldsymbol{v}} \leftarrow \mathtt{autodiff}\left(-\sum_{\boldsymbol{\theta}_i \sim r}\left[\tilde{w}_i \log \frac{p(\boldsymbol{y}|\boldsymbol{x}, \boldsymbol{\theta}_i)}{p(\boldsymbol{y}|\boldsymbol{v}, \boldsymbol{x}, \boldsymbol{\theta}_i)} + \frac{1}{S}\log \frac{p(\boldsymbol{y}|\boldsymbol{v}, \boldsymbol{x}, \boldsymbol{\theta}_i)p(\boldsymbol{\theta}_i)}{r(\boldsymbol{\theta}_i; \boldsymbol{\psi}^\star)}\right]\right)$

      s.t. $\boldsymbol{\psi}^\star = \arg\max_{\boldsymbol{\psi}} \frac{1}{S}\sum_{\boldsymbol{\theta}_i \sim r} \log \frac{p(\boldsymbol{\psi}|\boldsymbol{v}, \boldsymbol{x}, \boldsymbol{\theta}_i)p(\boldsymbol{\theta}_i)}{r(\boldsymbol{\theta}_i; \boldsymbol{\psi})}$

      $\boldsymbol{v} \leftarrow \max(\boldsymbol{v} - \gamma_t \hat{\nabla}_{\boldsymbol{v}}, 0) \quad \triangleright$ Take a projected stochastic gradient step

   **end for**

   **return** $v, \boldsymbol{\psi}^\star$

**end for**

At test time predict using $r(\boldsymbol{\theta}; \boldsymbol{\psi}^\star)$ and correcting via the importance weights $\tilde{\mathbf{w}}$.

---

**Algorithm 4** Black-Box Sparse VI [Batch] (Ours)

---

$\boldsymbol{\psi} \leftarrow \boldsymbol{\psi}_0 \quad \triangleright$ Initialize the variational parameters of the model

$\mathcal{I} \sim \mathsf{UnifSubset}\left([N], M\right) \quad \triangleright$ Get a minibatch of $M$ random indices of datapoints from the data

$\boldsymbol{v} \leftarrow \frac{N}{M}\mathbf{1}_{\mathcal{I}} \quad \triangleright$ Assign uniform weights and rescale likelihoods for invariance

**for** $t = 1, \ldots, T$ **do**

   $\mathcal{B} \sim \mathsf{UnifSubset}\left([N], B\right)$

   $\triangleright$ Use Algorithm 2 to obtain $\boldsymbol{\theta} \sim r(\boldsymbol{\theta}; \boldsymbol{\psi})$ and the corresponding importance weights $\tilde{w}$

   $\triangleright$ Compute the outer gradient wrt $\boldsymbol{v}$ using the gradient information of the inner optimization wrt $\boldsymbol{\psi}$

   $\hat{\nabla}_{\boldsymbol{v}} \leftarrow \mathtt{autodiff}\left(-\sum_{\boldsymbol{\theta}_i \sim r}\left[\tilde{w}_i \log \frac{p(\boldsymbol{y}|\boldsymbol{x}, \boldsymbol{\theta}_i)}{p(\boldsymbol{y}|\boldsymbol{v}, \boldsymbol{x}, \boldsymbol{\theta}_i)} + \frac{1}{S}\log \frac{p(\boldsymbol{y}|\boldsymbol{v}, \boldsymbol{x}, \boldsymbol{\theta}_i)p(\boldsymbol{\theta}_i)}{r(\boldsymbol{\theta}_i; \boldsymbol{\psi}^\star)}\right]\right)$

   s.t. $\boldsymbol{\psi}^\star = \arg\max_{\boldsymbol{\psi}} \frac{1}{S}\sum_{\boldsymbol{\theta}_i \sim r} \log \frac{p(\boldsymbol{\psi}|\boldsymbol{v}, \boldsymbol{x}, \boldsymbol{\theta}_i)p(\boldsymbol{\theta}_i)}{r(\boldsymbol{\theta}_i; \boldsymbol{\psi})}$

   $\boldsymbol{v} \leftarrow \max(\boldsymbol{v} - \gamma_t \hat{\nabla}_{\boldsymbol{v}}, 0) \quad \triangleright$ Take a projected stochastic gradient step

**end for**

**return** $v, \boldsymbol{\psi}^\star$

At test time predict use $r(\boldsymbol{\theta}; \boldsymbol{\psi}^\star)$ and correct via the importance weights $\tilde{\mathbf{w}}$.

---

---

**Algorithm 5** Black-Box Sparse VI [Batch] Pruning (Ours)

---

$\mathcal{C} := \{C_1, C_2, \ldots\}$ ▷ Set of coreset sizes in decreasing order for pruning rounds

$\mathcal{I}_1 \sim \mathsf{UnifSubset}\left([N], C_1\right)$ ▷ Get a minibatch of $M$ random indices of datapoints from the data

$\boldsymbol{v} \leftarrow \frac{N}{C_1}\mathbf{1}_{\mathcal{I}}$ ▷ Initialize to uniform weights and rescale for full-data evidence

**for** $C_i \in \mathcal{C}$ **do**

  $\mathcal{I} \sim \mathsf{Multi}(\boldsymbol{v}, C_i)$ ▷ Initialize the pruned coreset points via $C_i$ samples from current coreset

  $\boldsymbol{v} \leftarrow \frac{N}{C_i}\mathbf{1}_{\mathcal{I}}$ ▷ Reinitialize to uniform weights and rescale

  $\boldsymbol{\psi} \leftarrow \boldsymbol{\psi}_0$ ▷ (Re)initialize the variational parameters of the model

  **for** $t = 1, \ldots, T_i$ **do**

    $\mathcal{B} \sim \mathsf{UnifSubset}\left([N], B\right)$

    ▷ Compute the outer gradient wrt $\boldsymbol{v}$ using the gradient information of the inner optimization wrt $\boldsymbol{\psi}$

    $\hat{\nabla}_{\boldsymbol{v}} \leftarrow \mathtt{autodiff}\left(-\sum_{\boldsymbol{\theta}_i \sim r}\left[\tilde{w}_i \log \frac{p(\boldsymbol{y}|\boldsymbol{x}, \boldsymbol{\theta}_i)}{p(\boldsymbol{y}|\boldsymbol{v}, \boldsymbol{x}, \boldsymbol{\theta}_i)} + \frac{1}{S}\log \frac{p(\boldsymbol{y}|\boldsymbol{v}, \boldsymbol{x}, \boldsymbol{\theta}_i)p(\boldsymbol{\theta}_i)}{r(\boldsymbol{\theta}_i; \boldsymbol{\psi}^\star)}\right]\right)$

    s.t. $\boldsymbol{\psi}^\star = \underset{\boldsymbol{\psi}}{\arg\max}\frac{1}{S}\sum_{\boldsymbol{\theta}_i \sim r}\log \frac{p(\boldsymbol{\psi}|\boldsymbol{v}, \boldsymbol{x}, \boldsymbol{\theta}_i)p(\boldsymbol{\theta}_i)}{r(\boldsymbol{\theta}_i; \boldsymbol{\psi})}$

    $\boldsymbol{v} \leftarrow \max(\boldsymbol{v} - \gamma_t \hat{\nabla}_{\boldsymbol{v}}, 0)$ ▷ Take a projected stochastic gradient step

  **end for**

  **return** $\boldsymbol{v}, \boldsymbol{\psi}^\star$

**end for**

At test time predict using $r(\boldsymbol{\theta}; \boldsymbol{\psi}^\star)$ and correcting via the importance weights $\tilde{\mathbf{w}}$.

---

Table 4: Dataset statistics and hyperparameters for batch coresets used throughout experimental results of Section 4.

| | | | | | initial learning rate | | | | | | |
| Dataset | D | $N_{tr}$ | $N_{te}$ | $\psi$ | $u$ | $v$ | $\alpha$ | $z$ | $\sigma_{\psi_0}$ | Inner iters | Batch size |
|---|---|---|---|---|---|---|---|---|---|---|---|
| webspam | 128 | $100,948$ | $25,237$ | $10^{-3}$ | $10^{-4}$ | $10^{-3}$ | $10^{-3}$ | - | $10^{-6}$ | 100 | 256 |
| phishing | 11 | $8,844$ | $2,210$ | $10^{-3}$ | $10^{-3}$ | $10^{-3}$ | $10^{-3}$ | - | $10^{-6}$ | 100 | 256 |
| adult | 11 | $24,130$ | $6,032$ | $10^{-3}$ | $10^{-4}$ | $10^{-3}$ | $10^{-3}$ | - | $10^{-6}$ | 100 | 256 |
| half-moon | 2 | 800 | 200 | $10^{-4}$ | $10^{-2}$ | $10^{-1}$ | $10^{-3}$ | - | $10^{-4}$ | 50 | 128 |
| four-class | 2 | 800 | 200 | $10^{-4}$ | $10^{-2}$ | $10^{-1}$ | $10^{-3}$ | - | $10^{-3}$ | 50 | 128 |
| MNIST | $(1,28,28)$ | $60,000$ | $10,000$ | $10^{-3}$ | $10^{-2}$ | $10^{-2}$ | $10^{-3}$ | $10^{-2}$ | $10^{-10}$ | 20 | 256 |

Table 5: Hyperparameters for incremental coresets used throughout experimental results of Section 4.

| | | | initial learning rate | | | | | | |
| Dataset | $\psi$ | $\psi_{\text{Laplace}}$ | $v_{\text{Sparse VI}}$ | $v_{\text{BB Sparse VI}}$ | $\sigma_{\psi_0}$ | Inner iters | Outer iters | Batch size | MC samples |
|---|---|---|---|---|---|---|---|---|---|
| webspam | $10^{-3}$ | $10^{-1}$ | $10^{-1}$ | $10^{-1}$ | $10^{-3}$ | 200 | 400 | 512 | 64 |
| phishing | $10^{-1}$ | $10^{-1}$ | $10^{-2}$ | $10^{-2}$ | $10^{-6}$ | 50 | 200 | 256 | 64 |
| adult | $10^{-1}$ | $10^{-1}$ | $10^{-2}$ | $10^{-2}$ | $10^{-6}$ | 50 | 200 | 256 | 32 |

## C    Experiments details

In this section we provide additional details for our experimental setup, and further evaluation including experimentation over extended coreset size ranges, time plots and preliminary results with hypergradients as an alternative to iterative differentiation for bilevel optimization.

Note that to compute expectations for the posterior predictive of the model $f(\theta)$ on the test data under the variational family given by the intractable true coreset posterior $q(\theta|u,z)$, we utilize the variational program $q(\theta|u,z;\psi)$ which involves sampling from $r$ and correcting using the importance sampling scheme from Algorithm 2:

$$\mathbb{E}_{q(\theta|u,z)}[f(\theta)] \approx \sum_i \tilde{w}_i f(\theta_i), \quad \text{with} \quad \theta_i \sim r(\theta;\psi). \tag{18}$$

### C.1    Hyperparameters

For the experiments on batch coresets and baselines within the family of mean-field variational approximations presented in Section 4, we report the optimal mean predictive accuracy achieved, over independent runs of inference trials, with learning rates for model and coreset parameters taking values from $10^{-4}$ to $10^{-1}$ over logarithmic scale. We summarize dataset statistics, corresponding learning rates, along with the remaining selected hyperparameters (namely initialization of variance of our variational approximation, length of inner optimization loop and mini-batch size) for batch constructions of this set of experiments in Table 4. At our estimators we used 10 Monte Carlo samples from the coreset posteriors.

For incremental coresets we tuned the learning rates of the coreset weights $v$ to higher values: given that coreset evidence is initialised to 0 in Algorithms 1 and 3, we found empirically that, via allowing faster growth on the magnitude of the weight vector, the coreset construction can more easily escape local minima in the small size regime. Moreover, we allowed larger numbers of Monte Carlo samples: apart from being involved in the gradient estimation, in incremental coreset constructions this quantity defines the dimension of the geometry used at the greedy next point selection step, hence affording more samples can allow better selection of the coreset support. Larger learning rates on coreset parameters imply reduced stability over training iterations and a higher difficulty in hyperparameter tuning, hence demonstrate in practice the benefits attained by parameterisations that control the coreset evidence (see Section 3.2). We detail the choices for presented results in Table 5. We denote by $\psi_{\text{Laplace}}$ the learning rate used for fitting Laplace approximation on the logistic regression model (as done in the Sparse VI construction), as opposed to $\psi$ which is used when fitting a variational

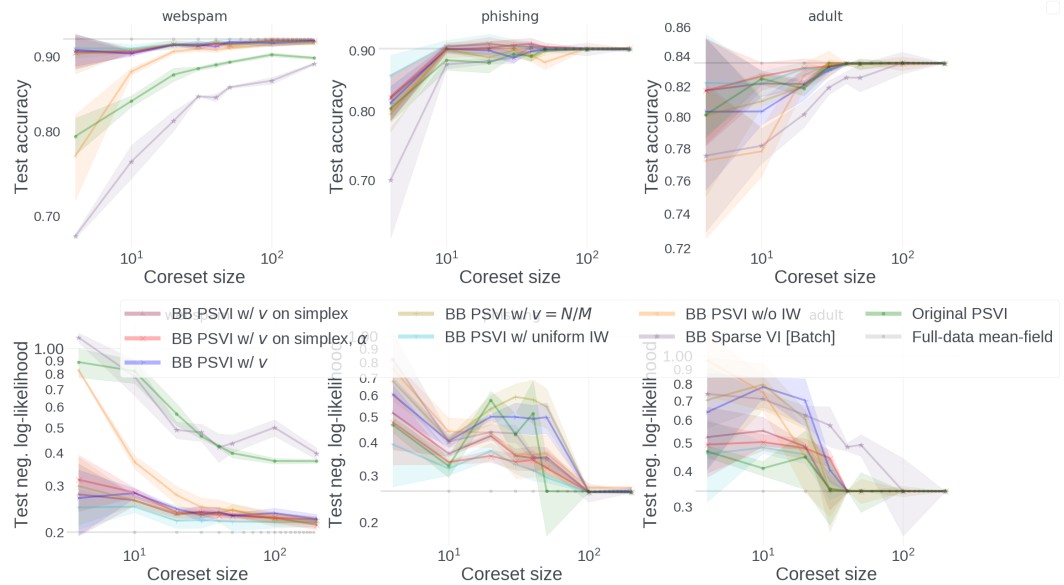

Figure 5: Predictive metrics across longer trials for the logistic regression experiment.

model on the coreset data (as done in BB Sparse VI); we also differentiate by indexing the learning rates for weights for the two Sparse VI constructions.

For the subset Laplace baseline we ran 500 gradient updates of a Laplace approximation with diagonal covariance, and used the Adam optimizer with learning rate $10^{-2}$, minibatches of size 256 and 32 Monte Carlo samples.

For Batch BB Sparse VI pruning we reset optimizers, and reinitialise the model and coreset parameters upon application of each pruning step. For the BNN experiment on the synthetic dataset we successively reduced in rounds from 250 initial datapoints to 100 and finally 20 datapoints, allowing training for 200 outer gradient updates before applying each pruning step.

In our bilevel problem, both for the nested and for the outer optimization, we use the Adam optimizer with the default hyperparameter setting of PyTorch implementation [40] and learning rates initialized per Table 4. For iterative differentiation we used the `higher` library [21]. For HMC we use the Pyro [5] implementation through TyXe [46]. Unless otherwise stated these hyperparameter choices will be followed in the additional experiments of this part. In this section, the labels of the pseudocoreset datapoints are always kept fixed.

## C.2 Computational resources

The entire set of experiments was executed on internal CPU and GPU clusters. For the logistic regression experiment, we used CPUs allocating two cores with a total of 20GB memory per inference method, while for the Bayesian neural networks we assigned each coreset trial to a single NVIDIA V100 32GB GPU.

## C.3 Logistic regression

**Predictive metrics** In Fig. 5 we present the predictive metrics we obtained across our full trials spanning coreset sizes from 4 to 200 on the 3 logistic regression datasets considered in Section 4.1. We note that for the incremental coresets we do not have direct control on the exact range of constructed coreset sizes over the experiment: At each trial we allocate a maximum number of next point selection steps for these methods, which however acts only as an upper bound of the attained largest coreset size (as an existing point might be reselected multiple times over the greedy next point addition steps). Indeed, for the Sparse VI construction [10] the coresets did not grow beyond 100 datapoints over the course of the experiment, and we did not include this method in the evaluation of this section. Moreover, for ease of visualization we removed baselines resulting in high

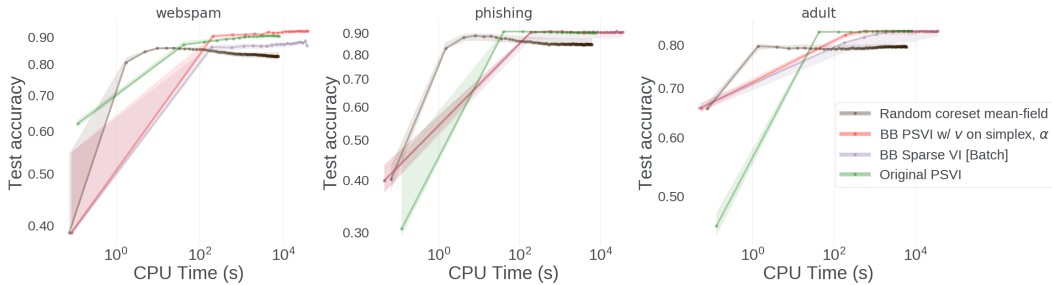

Figure 6: Test accuracy vs CPU time requirements with coreset size 100 for variational inference using a random coreset, PSVI, BB PSVI and BB Sparse VI Batch constructions at the logistic regression experiment.

variance, including the random coreset and BB-PSVI w/o rescaling. We can notice that the black-box constructions are capable to converge to the unconstrained mean-field posterior for sufficient coreset size, while convergence of the original PSVI construction might be limited by the heuristics employed over gradient computation (see `webspam` plot). Moreover, pseudocoreset methods achieve better approximation quality for small coreset size in high dimensions compared to Sparse VI and other true points selection methods.

**Computation time requirements**   In Fig. 6 we contrast the CPU time requirements for the execution of BB PSVI, BB Sparse VI, PSVI and the random coreset with $M = 100$ on our experiment, under usage of the same computational resources. Both pseudocoreset approaches solve a bilevel optimization problem, employing though starkly different machinery: BB PSVI relies on nested variational inference, while PSVI requires drawing samples from the true coreset posterior which are then used in estimating the analytical expression for the objective gradient. Hence, in the general case Monte Carlo sampling is required for asymptotically exact computation at every outer gradient step for PSVI, which makes it more expensive compared to our black-box construction. In practice a Laplace fit is used to approximate the coreset posterior; hence, making similar assumptions on the expressiveness of the coreset posterior (mean-field variational family vs Laplace with diagonal covariance), practical implementations of PSVI share the same order of time complexity with BB PSVI. However, BB PSVI is able to reach a higher peak (coresponding to the full data approximation of the selected variational family), as optimization is not hindered by amendments for intractability in the gradient expression. Finally, the gradient computation for BB PSVI via iterative differentiation— which was the optimizer of choice in this experiment—generally implies a higher memory footprint, as tracing the gradient of the inner optimization is required; this requirement can be alleviated via switching from a nested iterative differentiation optimizer to one that makes use of hypergradient approximations (see also next subsection).

## C.4   Bayesian Neural Networks

**Dynamics over the course of inference**   In Figs. 7 and 8 we present more instances extracted from the same experimental setting that was demonstrated at convergence (bottom rows) in Figure 3 of Section 4. We can discern the different dynamics in the optimization of pseudodata locations across our proposed constructions and initializations: Initialized on random noise, BB PSVI dynamics at the first stages separate the inducing points according to their categories; next, the inducing points are placed along the decision boundary, and more weight is assigned on data lying on critical locations of the empirical distributions corresponding to the different classes. Initialized on a random subset of the true data, BB PSVI convergence gets accelerated as the first two phases of move are not required. In the case of Batch BB Sparse VI, the locations of the coreset support are fixed, and gradually the scheme adjusts the weights to the importance of the data, prunes away the ones contributing redundant statistical information, and focuses on keeping data lying on the ends of the class distributions, that jointly specify the decision boundaries. Figure 9 demonstrates the evolution of our black-box variational objective for the coresets over a single trial. We notice that the objective presents more variance in the first stages of inference for random initialisation, as the noisy datapoints have to move significantly to discriminate the classes, however it can quickly reach the curve corresponding to the

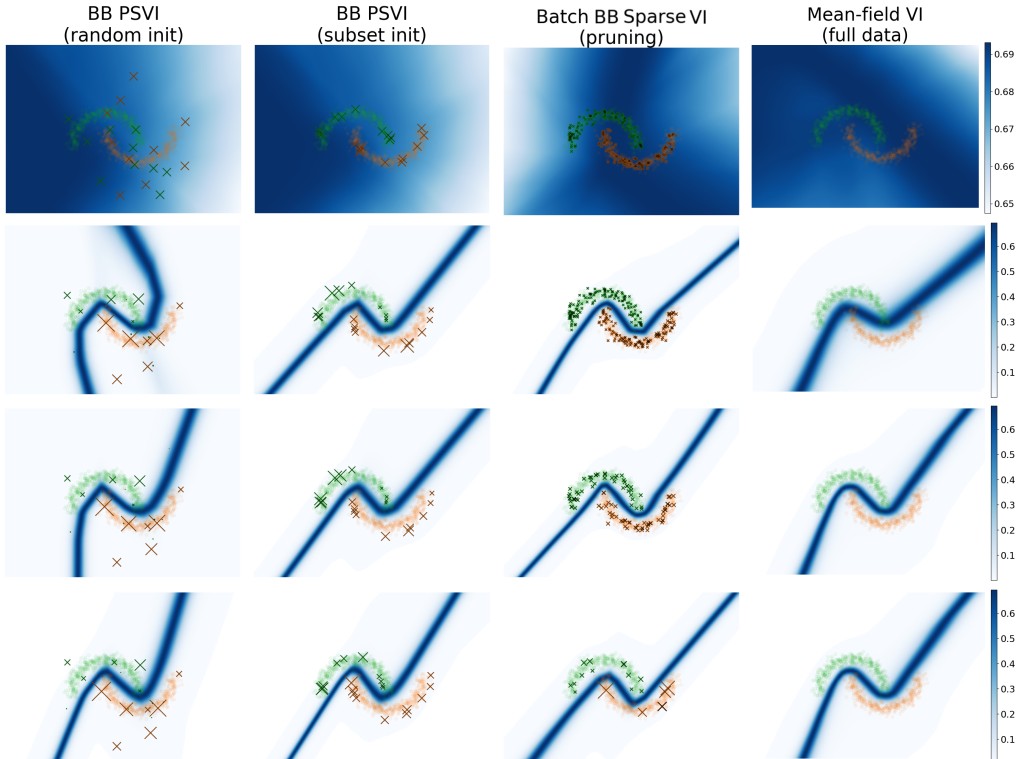

Figure 7: The different dynamics of coreset data optimization at four equidistant instances across the iterations of our inference trial for BB PSVI with 20 points sampled from a Gaussian, BB PSVI with initialization on a random subset of size 20, and Batch BB Sparse VI inducing sparsity via stepwise pruning $M = 250 \rightarrow 100 \rightarrow 20$ points. Mean-field VI on full-data is also plotted for reference.

subset initialisation. Regarding BB Sparse VI with pruning, we observe that, despite reinitializations of the network and the coreset point weights, the selection of good summarizing datapoints allows an overall increase of the variational objective, coping with the enforced shrinkage of the coreset size, and eventually achieving similar bounds with the coresets that use variational data.

**Justification of pruning strategy** Considering the interpretation of weights on the coreset datapoints as probabilities of a multinomial distribution that is learned on the data throughout inference in a way that preserves approximate Bayesian posterior computations (Section 3.2), allows us to justify our pruning step, where a larger coreset gets replaced by $K$ samples from this distribution. In this sense, pruning can be thought of as a means to introduce sparsity while minimizing diverging from the full-data posterior, as this operation corresponds to getting a small number of samples from the learned multinomial on the data, and retraining the model constrained on this sample.

## D   Continual learning

In Fig. 10 we apply black-box PSVI in a more challenging learning setting defined on the four class dataset, similarly to the synthetic data classification experiment presented in [24] for continually learning inducing points for Gaussian processes—a closely related concept to coresets. Instead of seeing training data from all 4 classes at once, we start with a 2-class classification problem, and, over 2 learning stages, the 3rd and 4th class get sequentially revealed. Whenever we move to the next classification problem, we:

- remove all $N_k$ true training data seen so far, and instead use $N_k$ samples *only* from the coreset support according to a multinomial defined via the vector $\boldsymbol{v}$ of coreset point weights,
- augment the coreset support with a new set of learnable datapoints from the most recently added class, initialising them at a total evidence proportional to the class size, and

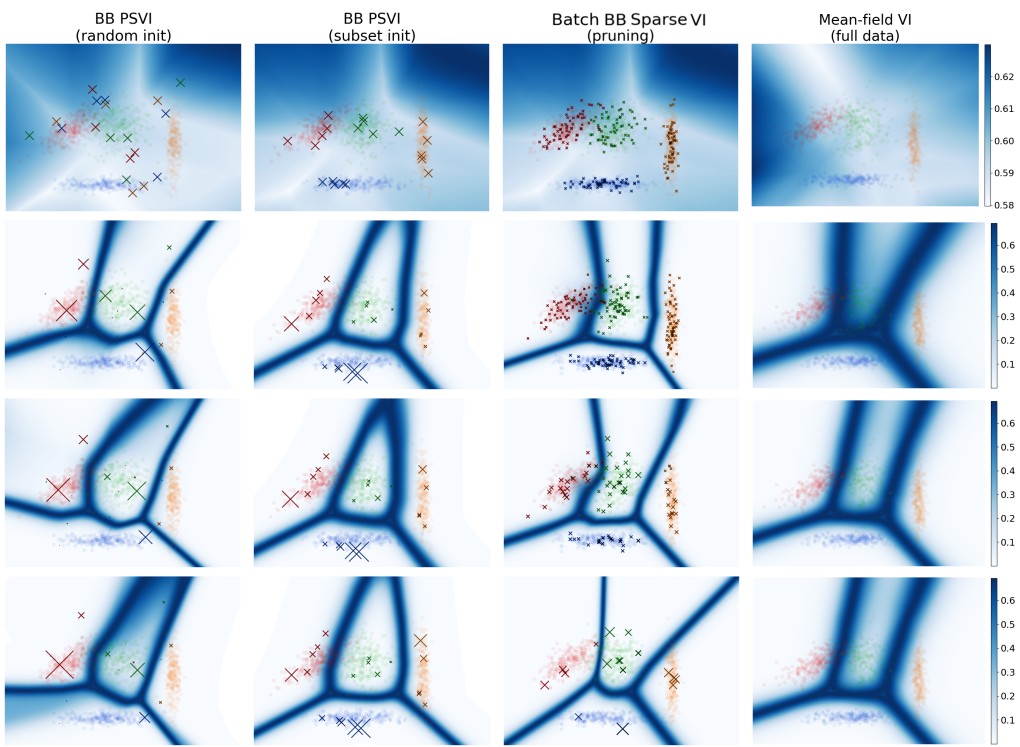

Figure 8: Counterpart of Fig. 7 with four snapsots over training for the multi-class dataset.

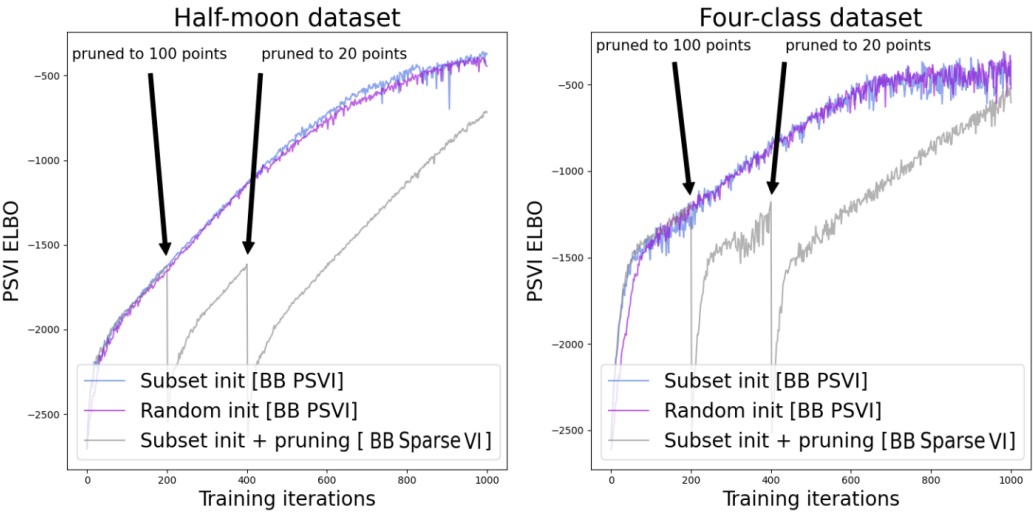

Figure 9: Variational objective vs outer gradient updates for the considered constructions of PSVI and Batch BB Sparse VI with pruning on the synthetic datasets. BB PSVI optimizes a batch of 20 pseudopoints since the beginning of training, while BB Sparse VI starts with a summary of 250 existing datapoints and gradually shrinks it to a batch of 20 informative datapoints after two rounds of pruning and retraining.

- reinitialise all variational parameters $\psi$ and adapt the last layer of our architecture to the number of classes of the new learning stage.

We can notice that the BB PSVI construction is able to successfully represent the historical training data, addressing the common issue of catastrophic forgetting, and provide representative posteriors throughout the 3 stages of our continual learning setting. Even though designed to summarise statistics

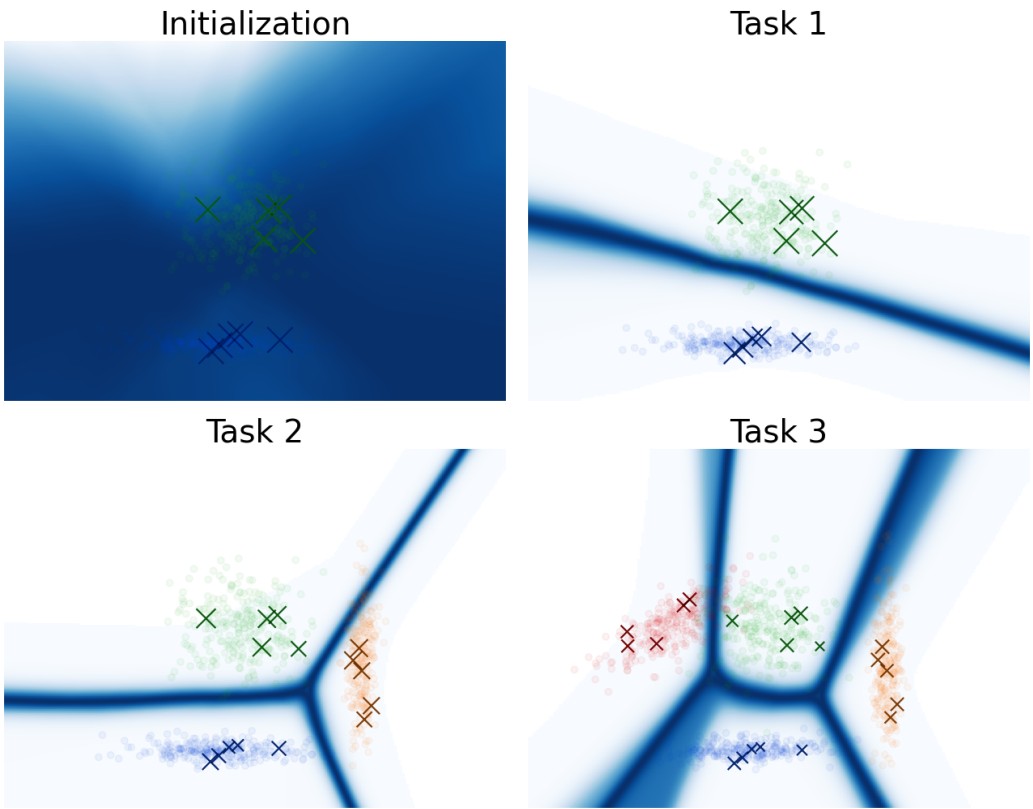

Figure 10: Continual learning setting: A coreset is constructed so that a BNN is fitted incrementally to the 3 classification tasks. We start with a coreset comprised of 10 points and over tasks 2 and 3 we increase coreset size to 15 and 20 datapoints respectively.

of earlier tasks, the coreset point locations and weights can be readapted to each new learning task, keeping the essential statistical information from the past.

## E   Monte Carlo inference on coresets

Extending on the introductory figure from the main text, in Fig. 11 we assess the results of running MCMC inference on the extracted variational coresets for a feedforward BNN trained on the halfmoon data. We visualise $100,000$ samples using Hamiltonian Monte Carlo after a warmup period of $20,000$ samples. We can notice that the variational coreset using BB PSVI is able to perform more intelligent selection of inducing point locations and weights, e.g. via placing increased importance on the ends of the halfmoon manifolds, hence being able to represent the original training dataset more compactly. This is reflected in an approximate posterior that is closer to the corresponding posterior computed on the full data; in contrast, random subsampling is prune to variance and often misses information about critical regions of the decision boundary. Overall, accelerating MCMC inference without losing much statistical information, as achieved via BB PSVI, can lead to better modeling of uncertainty on large scale data, compared to applying a fully variational treatment.

## F   Joint optimization of coreset support and variational parameters

In Fig. 12 we visualize the effects of jointly learning optimized coreset points locations and weights, along with variational parameters using the $\text{ELBO}_{\text{PSVI-IS-BB}}$ from Eq. (9). If we do not constrain the learning of variational parameters to the coreset points, these are unable to distill the relevant information from the true training data for our statistical model. As the joint optimization might be primarily driven by the true data, the coreset support might end up in non-representative locations

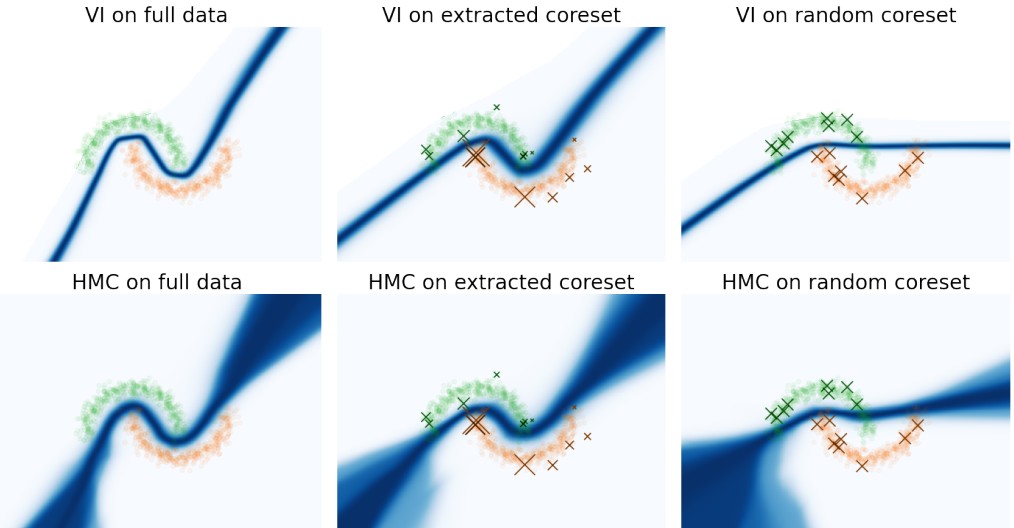

Figure 11: Posterior inference results via variational inference and Hamiltonian Monte Carlo on the full training dataset, a 16-points coreset learned via black-box PSVI, and a 16-points coreset comprised of uniform weights and datapoints selected uniformly at random. All inference methods use the halfmoon dataset and the same probabilistic model based on a feedforward BNN.

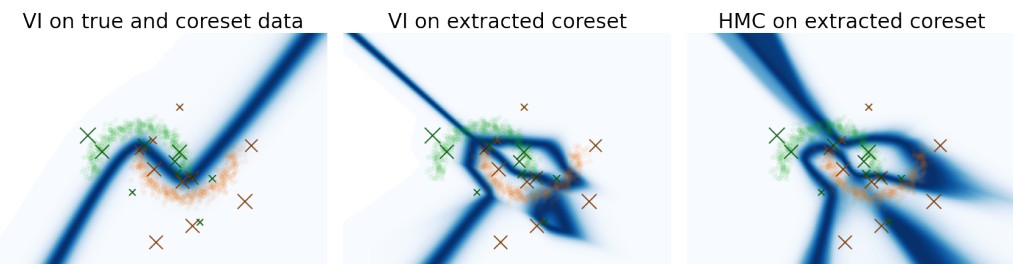

Figure 12: Effects of joint optimization on coreset posterior, using a feedforward BNN and a coreset with 20 datapoints on the halfmoon dataset. Blue shades represent predictive uncertainty, circles the original training data and crosses the coreset point locations with size proportional to the corresponding inferred weight.

of the data space, potentially resulting in incorrect decision boundaries after removing the original training datapoints as exhibited in this experiment. The design choices for HMC in this experiment are identical to Fig. 11.

## G    Importance sampling vs data dimensionality

In Fig. 13 we visualise the normalized effective sample size (ESS) using 10 Monte Carlo samples from our approximate posterior evaluated on the test data throughout PSVI inference. ESS takes consistently non-trivial values (larger than 0.1) even for 200-dimensional data, with a visible decreasing trend as data—and hence model—parameters dimensionality increases. Similarly to [12], the synthetic data for this experiment were generated using covariates $x_n \in \mathbb{R}^d$ sampled i.i.d. from $\mathcal{N}(0, I)$, and binary labels generated from the logistic likelihood with parameter $\theta = 5 \cdot \mathbf{1}_d$. We generated a total of $N = 1,000$ datapoints, with 20% of them used as test data, and constructed coresets of size $M = 20$.

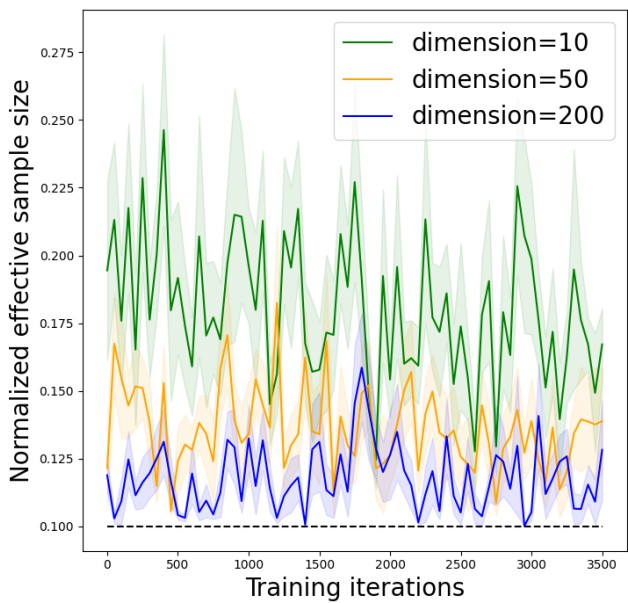

Figure 13: Normalized effective sample size (with standard errors) vs data dimensionality for inference using PSVI with 10 Monte Carlo samples in the Bayesian logistic regression model.

## H Visualization of MNIST summarizing data

In Fig. 14 we display the learned images and soft-labels for the coreset of size 30 constructed as part of the MNIST compression experiment. We can notice that the basic features of the original images are preserved in the pseudo-images, while the soft labels mainly assign larger score to the class of the corresponding image, and capture the uncertainty between similarly looking classes (e.g. 6 vs 8).

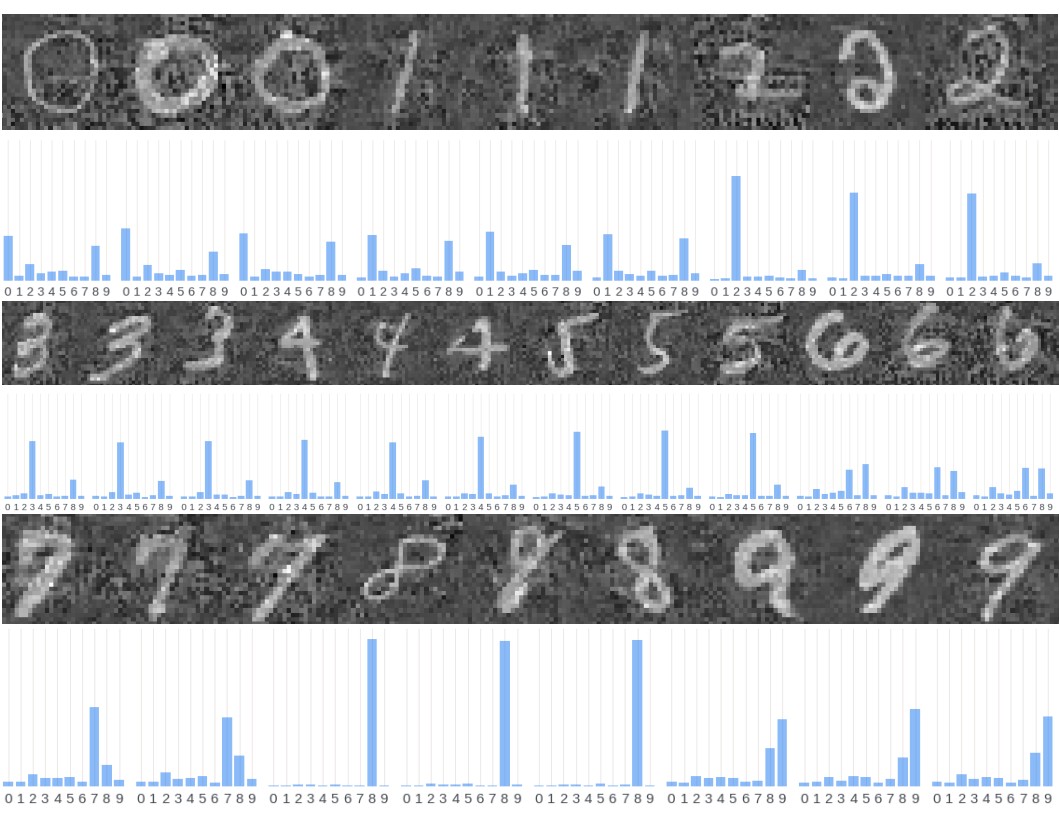

Figure 14: Visualization of the images and corresponding soft-labels for 30 pseudodata learned via black-box PSVI in the MNIST summarization experiment.