# OpenReview forum: "Black-box coreset variational inference"
_NeurIPS.cc/2022/Conference — NeurIPS 2022 Accept_

### Official Review · Reviewer_vKEG · 2022-06-28

**Rating:** 5
**Confidence:** 3
**Soundness:** 3 good
**Presentation:** 3 good
**Contribution:** 2 fair

**Summary:**

The authors present a new method to build coresets for inference tasks. That is, a method to build a "smaller" dataset so that the posterior over this new dataset is close to the true posterior. Their initial formulation follows the previously used approach finding the coreset so that the resulting distribution is closest (in KL divergence) to the true target. Since this in often intractable, the authors propose two approximations based on the use of a variational approximation (to approximate the coreset posterior) and importance sampling (to estimate expectations with respect to the coreset posterior).

**Questions:**

The most important point to be addressed is, in my opinion, W1. Simply put, convincingly show the benefits of using the resulting coreset over other naturally scalable approaches, such as plain VI. Addressing W2-W3 would be nice but not strictly necessary.

A few additional questions (these do not affect my score):

- Have you tried optimizing the full objective jointly, instead of using a nested optimization procedure?

- In table 1 the random coreset approach appears to work significantly better than Sparse VI. Do you have any intuition regarding why this may be? (I appears that the incremental/greedy approaches perform poorly, since Sparse BBVI also works worse than the random approach).

**Limitations:**

Are the limitations of the work explicitly discussed somewhere?

**Strengths And Weaknesses:**

Strengths:

S1: The method is presented in a clear way.

S2: The method is novel and simple to implement.

S3: The method appears to work well in the empirical evaluation presented.

Weaknesses:

W1: The proposed algorithm produces two outputs: (1) a coreset, and (2) a variational approximation (which approximates the coreset posterior, which in turn approximates the true target posterior). Regarding (2), a typical application of VI is as efficient (or even more efficient, since it is also compatible with subsampling and does not involve a nested optimization problem) and leads to similar or better results. Thus, the method's main benefit comes from producing a coreset (1), which may be used to enable other methods, such as efficient MCMC. But the paper does not explore this. I think it would be quite interesting to see how this method enables other algorithms (that would otherwise be very slow, e.g. MCMC) and how the resulting performance compares against plain VI.

W2: The use of a simple variational distribution may hurt the quality of the coreset built. I think this would happen because of two "forces" that may be pushing in different directions. The first term of the objective (eq. 9) is pushing towards a coreset that leads to a coreset-posterior that's close to the true posterior, while the second term to a coreset that leads to a coreset-posterior that is not too far from the variational approximation $q(\theta\vert \psi)$. A large increase in either of these terms may lead to an increase in the loss, and thus have to be balanced, possibly sacrificing coreset quality. Simply put, the use of a variational approximation leads to a tractable but biased approximation of the "ideal" objective to build the coreset from eq. 5, where the quality of the coreset may depend on the amount of bias. While the use of importance sampling as a debiasing tool as presented in the paper may ameliorate this issue, importance sampling is known to work poorly in high dimensions. I think that adding a more explicit discussion about this bias should be included somewhere in the paper.

W3: Related to W2, another way to improve the bias issue would involve the use of a more powerful posterior (e.g. full-covariance Gaussian, flows). It would be interesting to see how the performance of the method improves in this case, and if the use of the resulting coreset leads to better performance when used with other methods, such as MCMC (related to W1).

I think W1 is quite important, specially since VI is already giving an approximate solution. As I see it, a motivation for building coresets is that they may enable more accurate but otherwise more inefficient methods, such as MCMC. Thus, exploring whether the use of the resulting coresets with more accurate methods (e.g. MCMC) yield better results than the plain VI solution seems particularly relevant. (This is also related to W2, as the quality of the variational approximation may affect the quality of the coreset obtained.)


A few additional comments:
- Equation 4 defines $q(\theta\vert u,z) = p(\theta\vert u,z)$. But then in equation 5 both $p$ and $q$ are used. I think it would be better to just decide to use one of them, and stick to it throughout.
- I think the paragraph between lines 97-102 may need some editing? It seems that the line starting from the textbf second is redundant with what is said before?

---

> ### Author Response · Authors · 2022-08-02
> **Response to the review**
>
> Thank you very much for your detailed and thoughtful review. You bring up various interesting points, so we are glad to see that you are all in all supportive of our paper. We will address your concerns below:
>
>
> > W1: run MCMC on the coreset and compare to full dataset VI
>
> Thank you for suggesting this additional experiment to highlight the utility that coresets can provide. We had taken the motivation for extracting coresets somewhat for granted due to the research area being rather active, but explicitly highlighting use cases like this makes the paper more accessible to a general audience of course.
>
> We have included some plots based on the halfmoon dataset in Appendix E.2. We compare inference via HMC on a coreset learned using blackbox PSVI, a coreset built on a random subset, and the full dataset. We observe that running HMC on the PSVI coreset brings the approximate posterior close to the one obtained via HMC on the full data, while the non-representative data selection of the random coreset is unable to reproduce the true decision boundary. As HMC provides better uncertainty estimates compared to VI on the full dataset, scaling MCMC methods via coreset compression to larger Bayesian  neural network architectures or other black-box models is indeed a compelling direction for future research.
>
> However, we would like to strongly emphasize that coresets are not limited to running expensive inference methods on them that would not scale to the full dataset. Instead, there are various downstream use cases and further methods to be researched in settings such as continual learning and other approaches necessitating dataset summarization  – please see the general response for an outline of potential follow-up work. Black-box models, such as neural networks, are arguably the most commonly used type of model in practice and often plagued by issues such as forgetting. So we view opening the doors for developing principled black-box Bayesian inference methods based on coresets for these models, and any other model one might want to consider, as the core contribution of our paper. This has the potential of attracting significant additional interest in Bayesian coreset methods from the wider NeurIPS community.
>
>
> > W2: lack of discussion around the bias resulting from approximate inference
>
> Thank you for pointing out that you would have liked to see more discussion around this. We completely agree with your points around the different forces in our coreset objective as well as the possible impact of the variational family. We will make use of the additional page allowed in the camera-ready paper to add a discussion around these points and investigate more expressive variational families and their impact on the learned pseudodata alongside inference on them (see discussion around W3).
>
>
> > W3: do more expressive approximate posteriors lead to better coresets?
>
> This is an interesting suggestion that we will be keen to further investigate, e.g. by using full-covariance Gaussians or normalizing flows on small-scale experiments such as the HMC experiment we provided in response to W1. For larger scale use cases we’d like to highlight that structured posteriors for BNNs are an active area of research in their own right and would require a significant investment to scale more expressive variational families.
>
>
> > Have you tried optimizing the full objective jointly, instead of using a nested optimization procedure?
>
> We indeed have explored this approach, however this setup leaks information from the full dataset into the parameters for the approximate posterior over the model parameters. While this leads to an identical approximate posterior to running VI on the full dataset, the resulting coresets are effectively ignored by the optimizer and do not represent the data. We have added Figure 12 to Appendix E.3 alongside a short discussion.
>
>
> > In table 1 the random coreset approach appears to work significantly better than Sparse VI. Do you have any intuition regarding why this may be?
>
> This seems to be an issue in particular on the webspam dataset. We hypothesize that this is due to the greedy selection procedure running into local optima (as you also seem to suggest). We will revisit these particular results for the camera-ready paper.
>
>
> Thank you also for your additional comments, we have incorporated them into the revised version of the paper that we uploaded alongside the rebuttal. We hope our additional MCMC experiments as well as our comments have been sufficient to alleviate your concerns and that you will be confident to recommend acceptance more assertively by raising your score. We are of course happy to engage in further discussion should you require additional details or clarifications.

---

> > ### Comment · Reviewer_vKEG · 2022-08-05
> > **Thanks for the reply**
> >
> > Dear authors,
> >
> > Thanks for your reply, and presentation about potential applications of coresets. Despite the detailed answer, I'm inclined to keeping my score. I have concerns about the interplay between the variational objective and the quality of the coreset built, as explained in the review (importance sampling, while useful, won't help much in high dims). Additionally, the empirical evaluation is focused on tasks where one could simply run mean-field VI (which is done in the work), which is as efficient (if not more) than the method. While some additional experiments were added to the supplementary, they were mostly done on very simple cases and do not compare to other coresets approaches. I believe that a thorough empirical evaluation targeting these aspects could really strengthen the paper and showcase the methods strength (if it still performs significantly better than relevant baselines).

---

> > > ### Author Response · Authors · 2022-08-05
> > > **Thank you for engaging in the discussion**
> > >
> > > Dear reviewer, thank you for promptly engaging in the discussion, we highly appreciate being given the opportunity to respond to you prior to the end of the discussion period.
> > >
> > > As you can probably imagine, we are disappointed and a little bit surprised that your opinion of the paper remains unchanged. Your review had clearly requested a demonstration of the advantage of our coreset learning method over VI on the full dataset to resolve the core remaining concern about our work. We believe that we have addressed this convincingly, showing that MCMC inference on the coreset matches the predictive distribution of MCMC on the full dataset substantially more closely than VI. Of course our experiment is limited to a toy dataset, so we are happy to further substantiate these qualitative results with more quantitative ones complementing the logistic regression benchmarks for the camera-ready version. As you surely understand we had to prioritize with the limited time available and chose experiments that are fast to execute and fit within our existing codebase in order to allow us to address the different questions that other reviewers had asked as well.
> > >
> > > We would like to reiterate that our paper presents a principled and modular extension of Bayesian coresets to black-box models. You are absolutely right to question whether importance sampling will be the ultimate answer for coreset learning in high dimensional models. It may well be the case that Sequential Monte Carlo and more advanced methods such as those pointed out by R`etFX` will be better able to yield higher effective sample sizes, taking advantage of higher quality proposal distributions in the form of structured posterior approximations. However, developing scalable yet expressive approaches for inference in black-box models with potentially complicated posterior shapes and high-dimensional parameter spaces is a long-standing, yet unsolved, problem among the Bayesian machine learning community. Our method is agnostic to the variational posterior used and we purposefully focused our empirical studies on the most widely-used black-box models and inference methods. Studying more accurate approximate inference on the coreset, different sampling techniques and the interplay between the two is a highly non-trivial research direction in its own right and not merely a tangential issue of our present work.
> > >
> > > This is precisely why we strongly believe that our paper is of interest to the community: it represents a **significant advance** for Bayesian coreset methods in supporting arbitrary black-box models for the first time, while offering a wide range of exciting opportunities for future work building on top of ours and further improving the different components of our pipeline. For bespoke high-dimensional models, the avid reader can replace our importance sampling outer-loop with more elaborate samplers paired with adequate proposal/variational distributions. The pattern we set out for how to modularize and make this objective function black-box remains intact.
> > > We will add a paragraph about this to our discussion, to clarify the modular structure of variational families and samplers to generate useful samples for evaluating our ELBO.

---

### Official Review · Reviewer_iNe6 · 2022-07-12

**Rating:** 5
**Confidence:** 3
**Soundness:** 3 good
**Presentation:** 2 fair
**Contribution:** 3 good

**Summary:**

The paper proposes a general pseudo-data sparse black-box variational inference method. Using the idea of Bayesian pseud-ocoresets from [25], the paper proposes the ELBO formulation of the sparse pseudo-data variational inference. Importance sampling is used for the variational distribution. A nested variational inference objective is further developed over the pseudo points and the variational parameters. How to update or select the coreset data is also discussed. Experiments are carried out on two different models: logistic regression and Bayesian neural networks.

**Questions:**

1. I would hope to get more explanation on why importance sampling is used instead of directly using ELBO defined by the approximate posterior ($q(\theta|u,z;\phi)$).
2. Also it would be more convincing if there is some experiment that shows the promises of using this approach to achieve something that other methods cannot.

**Limitations:**

N.A.

**Strengths And Weaknesses:**

Strengths:

- The proposed black-box pseudo-coreset variational inference formulation is novel. And the derivation follows through.

Weaknesses:

- The experiment results on MNIST do not seem very competitive. With 10 data points in the coreset, the inducing point type of approaches can achieve 90+% accuracy. Also, there is some discrepancy with the reported accuracy of Dataset Condens.[44] in the original paper. Also, for Dataset Condens.[44], SLDD[37], and Dataset Distill.[42] the model uses 100 condensed data to train NN from scratch and achieve the reported accuracy. While for the proposed method, it is also using the labels of all training data for learning $\phi$, $u$, and $z$.
- The motivation behind importance sampling is not very clear to me. It does not exactly explain why not to use the ELBO defined by the approximate posterior ($q(\theta|u,z;\phi)$) directly.
- For the regression tasks and MNIST, it would be more interesting to look at the pseudo-dataset learned and analyze if they make sense, just as what is done with the simulated datasets.

---

> ### Author Response · Authors · 2022-08-02
> **Response to the review**
>
> Thank you very much for your positive review. We think you have fairly summarized the contribution of our paper, but that there are a couple of minor misunderstandings remaining. We will clarify these along with your questions below:
>
> > The motivation behind importance sampling is not very clear to me.
>
> The need for importance sampling arises from not having access to the true posterior over the model parameters given the coreset, but having to resort to approximate inference. In order to learn a coreset, we need to draw samples from the variational family given by the posterior over coresets $p(\theta| u, z, v)$ and score them in the ELBO, so substituting in an approximate posterior $q(\theta)$ to sample from would result in bias. We leverage importance sampling in order to remove this bias, as illustrated in our method section 3 as one of the two key technical problems of the original PSVI criterion. As also discussed in the response to R`etFX`, we can only fully remove the bias in the asymptotic limit, but even with a finite number of importance samples we obtain a tighter bound on the model evidence.
>
>
> > it would be more convincing if there is some experiment that shows the promises of using this approach to achieve something that other methods cannot.
>
> Such experiments are already present in the paper. The Bayesian neural network in section 4.2 is a non-tractable model, to which PSVI could not and has not previously been applied. All baselines in Table 3 are non-Bayesian methods.
>
> Strictly speaking, the logistic regression model in 4.1 also does not have a closed form posterior, however PSVI works around this with a Laplace approximation. However, PSVI does not correct for the approximate nature of its posterior and as such has a disconnect between the practical implementation and what the ELBO in PSVI calls for: samples from $p(\theta| u, z, v)$. We show consistent improvements in predictive performance with our principled approach that is built from the ground up with approximate inference in mind, in particular with small coresets, and make the objective end-to-end learnable.
>
>
> > The experiment results on MNIST do not seem very competitive. (...) there is some discrepancy with the reported accuracy of Dataset Condens [44] in the original paper. Also, for Dataset Condens.[44], SLDD[37], and Dataset Distill.[42] the model uses 100 condensed data to train NN from scratch and achieve the reported accuracy.
>
> In the MNIST experiment we applied our inference framework on BNNs using the LeNet architecture. Hence, for consistency the results for all baselines in Table 3 (including Dataset Condensation [44]) correspond to the reported performance of the discussed training methods for the LeNet architecture. Regarding the number of learnable parameters: SLDD[37] also optimizes soft labels. Moreover, although not consistently outperforming all baselines, our method is the first dataset condensation technique that can be used for Bayesian inference, enhancing predictions with uncertainty quantification.
>
>
> > For the regression tasks and MNIST, it would be more interesting to look at the pseudo-dataset learned and analyze if they make sense.
>
> We have added a visualization of the trained MNIST coresets in Appendix E.5, Figure 14. For the logistic regression benchmarks, this is of course less straight-forward than for image data. We will investigate using e.g. intervals or kernel densities of marginal features and comparing those with the coresets in the camera-ready paper.
>
> We hope we have been able to address your concerns and that you feel comfortable to recommend acceptance decisively by increasing your score. We are of course happy to engage in further discussion.

---

> > ### Comment · Reviewer_iNe6 · 2022-08-08
> > **Re: Response to the review**
> >
> > Thanks very much for the clarifications. I feel the paper would be much stronger if the experiments can include motivation examples that you mention in your general response, such as continual learning or transfer learning. Therefore I will keep my current score and inclining to acceptance.

---

> > > ### Author Response · Authors · 2022-08-08
> > > **We added a continual learning experiment for the rebuttal**
> > >
> > > Dear reviewer,
> > >
> > > Thank you for your response. We completely agree that showing a continual learning result would be valuable, and thus had added one for our rebuttal we shared last week.
> > >
> > > Please kindly have a look at our rebuttal, we updated the appendix with a continual learning experiment when we sent the rebuttal, as we noted it is one of the examples from the VAR-GP continual learning paper by Kapoor et al. from ICML 2021 that we replicated using our core sets with BNNs instead of Gaussian Processes.
> > >
> > > The description for continual learning we added in the text thus has an experimental counterpart in our supplement, we agree that it would be valuable to work it into the main paper and will do so, but until then please view the supplement and our description for the result.
> > >
> > > We hope this convinces you to raise your score and recommend acceptance more emphatically.

---

> > > > ### Comment · Reviewer_iNe6 · 2022-08-08
> > > > **Re: We added a continual learning experiment for the rebuttal**
> > > >
> > > > Thanks for adding the continual learning experiment. It is a nice starting example. As you mentioned, the current experiment is very minimal and for demonstration only. I think it would be more interesting to compare with standard approaches, to show that with the coresets it is able to learn efficiently and effectively (for example under domain shift). Also, it would be much more convincing if it is performed on a real-world problem that is not easy to learn. In my opinion, demonstrating the utility of coresets in this kind of setting is very important for the paper. I would keep my current rating for now.

---

> > > > > ### Author Response · Authors · 2022-08-09
> > > > > **Our paper’s topic is on the estimator for Bayesian Coresets, not CL**
> > > > >
> > > > > Dear reviewer,
> > > > >
> > > > > Thank you, we are  somewhat disappointed our rebuttal with multiple new experiments did not sway you positively but thank you for engaging and providing more feedback.
> > > > >
> > > > > We do want to point out, however, that what you are asking for is a full continual learning paper which is likely a whole separate work and would create challenges in scientific dissemination to be compressed into one manuscript together with the estimator and all the technical details we demonstrate on core sets here, including experiments showing that the estimator works on a single task.
> > > > >
> > > > > Our core contribution is the technical advance to make PSVI and Sparse VI-like coresets run with a principled black box estimator that overcomes specific technical challenges of the older formulations to facilitate models like BNNs and to elucidate that with quantitative and qualitative examples to show the versatility and quality of the method, thereby going beyond previous Bayesian coresets publications, which we do both empirically and theoretically.
> > > > >
> > > > > The continual learning example with BNNs and these coresets is just an extra application to demonstrate the utility and flexibility of our proposed estimator, and the example follows a recently published paper on Bayesian Continual learning (from ICML 2021) and is not trivial since it captures uncertainty.
> > > > > We generalized an empirical result in Bayesian incremental/continual learning previously attainable with Gaussian Processes as suggested by Kapoor, Karaletsos, and Bui at ICML 2021 to BNNs here by a straightforward application of our method as a side result. We find that pretty cool as one of many experiments, we hope you share our enthusiasm.
> > > > >
> > > > > We will also consider your advice to write a full-blown CL work based on this method and are excited about your belief in the applicability of the approach to warrant that suggestion.
> > > > >
> > > > > Thank you again for your feedback and your generally positive assessment of the paper.

---

### Official Review · Reviewer_etFX · 2022-07-13

**Rating:** 7
**Confidence:** 4
**Soundness:** 3 good
**Presentation:** 3 good
**Contribution:** 3 good

**Summary:**

The authors propose a variational inference method (BB PSVI) based on PSVI, which learns a variational approximation over model parameters given pseudo data that approximates the true posterior over model parameters given real data by maximizing a variational lower on the true data likelihood (ELBO-PSVI).
A valid but generally intractable choice for the variational approximation is the true posterior over model-parameters given pseudo-data. To work around the tractability issues the authors propose to perform (adaptive) importance sampling which leverages another variational proposal over model-parameters by maximizing a lower bound on the pseudo-data likelihood (ELBO-IP).
The authors show that replacing the pseudo-data likelihood term in the PSVI-IS ELBO with the IP ELBO results again in a lower bound on the data likelihood. This variational lower bound (ELBO-PSVI-IS-BB) constitutes their final objective used to optimize the pseudo-observations. The variational approximation over model-parameters given pseudo-data can be modified to incorporate an weighting factor for individual pseudo-datapoint that can be learned jointly by maximizing the PSVI-IS-BB ELBO.


**Questions:**

The authors observe that existing approaches are limited by the data dimensionality while their approach allows for accurate approximate inference on the full data set regardless of dimensionality. While seemingly true for the presented models and datasets, intuitively the quality of the importance sampling estimate scales badly with dimensionality and is heavily reliant on good proposals for the position of the pseudo data points. The mechanisms to acquire positional proposals (pruning, incremental selection, or even potentially smarter ones like Bayesian optimization) seem to also scale badly with dimensionality as exponentially more positions have to be considered as the dimensionality increases. Is this an issue you'd expect to see in practice on different datasets? Why isn't this an issue for the webspam dataset (d=128)? Can you please elaborate a bit on that topic.

**Some minor things I noticed:**
- Line 64-66: the posterior distributions $p(\theta \mid u ,z, v)$ and $p(\theta \mid x, y)$ are over model parameters and not over the pseudo-dataset and dataset as stated in the text.


**Limitations:**

See questions regarding scaling with dimensionality

**Strengths And Weaknesses:**

**Originality:**
The idea of using an importance sampling based approximation of the *pseudo posterior* to derive a black box version of PSVI is original and methodologically interesting and allows the use of PSVI for previously intractable black box models. As such I believe that this is a potentially interesting paper for both methodologically interested readers and application-oriented ML partitioners alike.

**Quality:**
The methodology and technical argument are sound.

**Clarity:**
The paper is well structured and clearly written. Given that the authors are using a nested variational inference step using self-normalized importance sampling, they should definitely cite recent work on Nested Variational Inference [1] in their related work subsection discussion Variational inference, importance weighting, and Monte Carlo objectives.

[1] Zimmermann, H., Wu, H., Esmaeili, B., & van de Meent, J.-W. (2021, December). Nested Variational Inference. 35th Conference on Neural Information Processing Systems (NeurIPS 2021).

**Significance:**
The presented results show significant improvements in accuracy and test log-likelihood for the presented models and datasets. I am however skeptical how the proposed approach performs on data with complex structure and high intrinsic dimensionality as the importance sampling approach is likely to fail here regardless of smart position-initialization and acquisition schemes. This trend seems to be already visible in the evaluation of the presented datasets (webspam, phishing, adult) where the effective sample size is non-trivial but rather low. I believe that additional discussion and an additional experiment on synthetic data which systematically evaluates the performance of the proposed method under increasing dimensionality would be great to gain further insights.

---

> ### Author Response · Authors · 2022-08-02
> **Response to the review**
>
> Thank you very much for your encouraging and supportive review. We will address your questions and concerns below:
>
> > Given that the authors are using a nested variational inference step using self-normalized importance sampling, they should definitely cite recent work on Nested Variational Inference
>
> Thank you for making us aware of this reference, it is clearly relevant to our paper. We are now citing it in the related work section.
>
>
> > I am however skeptical how the proposed approach performs on data with complex structure and high intrinsic dimensionality as the importance sampling approach is likely to fail here regardless of smart position-initialization and acquisition schemes
>
> We emphasize here that the importance sampling applies to the model parameters, not the coresets. So the dimensionality to consider is that of the model, not the data. While in logistic regression these are of course the same, this is typically not the case in black-box models. So the importance sampling of model parameters and optimization of pseudo-data are two distinct, but of course challenging issues as you rightly observe.
>
> To add some extra insights related to the scaling of our importance sampling scheme with model dimensionality, we added a related experiment in Appendix E.4. We simulated binary classification data over increasing dimensionality using the same generative process. In the corresponding plot the effective sample size indeed reduces with increasing dimensionality, which indicates the difficulty of obtaining good proposals for small coreset sizes in high dimensions — however, even for 200-dimensional data this metric remains non-trivial, indicating consistent gains for the importance sampling scheme.
>
> The dimensionality of the model parameters is indeed often large for black-box models such as Bayesian neural networks. In our view, importance sampling offers two compelling advantages: (i) it makes our method asymptotically exact by correcting for the bias resulting from using an approximate posterior over the coreset. This bias had not been accounted for in previous work on Sparse VI [8] and PSVI [25], which had simply hoped that the approximate posterior would be a close enough fit. (ii) Even with a finite number of importance samples, the lower bound on the evidence is strictly tighter than the Monte Carlo ELBO, see e.g. work on [Importance Weighted Autoencoders](https://arxiv.org/abs/1509.00519) (Burda et al., ICLR 2016)[7]. In simple terms: importance sampling can only reduce bias, even with a finite number of samples.
>
> However, there may of course be a trade-off for large-scale models under a limited compute budget, where more biased gradient updates may be preferable to fewer updates with a smaller bias. Exploring these trade-offs and limits empirically will certainly be an important avenue for future work, however we do want to highlight that previous works on Bayesian coresets have primarily focused on logistic regression on relatively small scale datasets with the assumption of explicit and tractable closed form gradients, so our work represents a significant step up in scale by moving towards neural network models.
>
> We hope we have been able to resolve any remaining doubts about our paper and that as the highest scoring reviewer you will be championing our paper. We are of course more than happy to engage in further discussion should you require any additional clarifications.

---

> > ### Comment · Reviewer_etFX · 2022-08-09
> > **Thanks for the reply**
> >
> > Thank you for the reply and your extensive effort to address our concerns. I think that the additional experiment strengthens the paper in two ways:  (1) it demonstrates the limitations of important sampling intrinsic to the method (which I believe is important to explicitly discuss in the paper) and (2) it shows that non-trivial effective sample sizes can be achieved for medium to high dimensional data (in the case of logistic regression).
> >
> > More generally, I believe extracting core-sets is an interesting area of research that goes beyond providing summarized surrogate data for expensive inference algorithms and that the authors propose a novel and conceptually interesting framework. The method itself has its limitations, e.g. application in high dimensions, but so do most methods and I do believe that the revised experiments, while not an extensive case study (not unusual for a methodologically oriented paper), demonstrate the utility of the method. There also seems to be a path towards addressing/mitigating these issues by integrating more sophisticated techniques like SMC in future work. I believe that a slightly revised manuscript that highlights the motivation and limitations of the method (but also core-set methods in general) and integrates the new experiments would make a good addition to the conference. In line with that I will increase my score to an Accept.

---

> > > ### Author Response · Authors · 2022-08-09
> > > **Thanks for your response**
> > >
> > > Thank you for your response as well as your succinct summary of the contribution and the opportunities for future work in our paper. We are pleased to see that we have been able to address your core concerns with our additional experiments and that you would like to see our work featured at the conference. We will of course incorporate these new results along with a more detailed discussion of the motivation and limitations of our method specifically and coresets in general as suggested by the reviewers into the next revision of our manuscript.

---

### Author Response · Authors · 2022-08-02
**General response: coresets have downstream potential beyond scaling expensive inference**

We thank the reviewers for their thoughtful comments and feedback. We are delighted to see a consensus for acceptance of our paper. The reviewers describe the work as “original and methodologically interesting” (R`etFX`) and the methods presented as “novel and simple to implement” (R`vKEG`), so we are pleased to see our generalization of Bayesian coresets to black-box inference for probabilistic models to be well-received.

We are, however, slightly concerned that the contribution of our paper has not fully come across with two of the reviewers only rating this aspect as “fair”. We believe that our work opens up the development of further coreset-based Bayesian methods in a wide range of areas and models. While e.g. R`vKEG` correctly notes that one possible (and arguably the original) motivation for coresets is running more costly but accurate inference methods on the reduced dataset, coresets have the potential to address a wide range of issues encountered especially in black-box models not admitting exact inference. These are precisely the models for which our work enables the use of coresets, as demonstrated here for Bayesian neural networks.

One such example is that of catastrophic forgetting in neural networks, which has attracted significant attention from the community. The predominant weight space inference methods, e.g. [Variational Continual Learning](https://arxiv.org/abs/1710.10628) (Nguyen et al., ICLR 2018), store knowledge about past tasks in the approximate posterior over the parameters, which loses information due to its inexact nature. Coresets, on the other hand, allow for learning pseudodata to compactly represent the information in past datasets. In the recent work [VAR-GP: Variational Auto-Regressive Gaussian Processes for Continual Learning](https://arxiv.org/abs/2006.05468) (Kapoor et al., ICML 2021) inducing points, a concept tightly related to coresets, were utilized in the context of GPs to enable continual learning. But how can one generalize the idea of coresets or inducing points to arbitrary models that do not necessarily have as readily attainable general posterior representations in terms of a compressed dataset as kernel-based techniques such as GPs to enable similar approaches for other models? To this end, we illustrate the utility of our approach to enable such inference for arbitrary models and provide a conceptual visualization in Appendix E.1 from an additional experiment with the four-classes toy dataset, where we learn the third and fourth class sequentially, only carrying the learned coresets forward to prevent forgetting. Please see our separate comment for the detailed setup.

This is of course just a sketch for a coreset-based continual learning method to motivate an application that requires dataset summarization apart from enabling scalable posterior inference. We would require further experimentation to formulate a competitive method which ties successive tasks together and anticipate needing to leverage the uncertainty over the parameters to solve more challenging benchmarks, but hopefully we have given a flavor for the type of approaches that could be developed based on our paper even when using our current estimator without any bespoke methodology for continual learning. We would expect similar opportunities for extracting privacy-preserving representations of datasets, a significant concern for many machine learning practitioners these days.

Another contribution of our work is that we provide a principled, unified formulation of Sparse VI [8] and PSVI [25] for black-box models. This provides practitioners with the option of avoiding the potentially difficult high-dimensional optimization of the pseudodata by blocking gradients to those variational parameters, only selecting them from existing data in a batched or iterative manner.

Further, our work is deeply connected to various active areas of research. The methods we propose in the paper could immediately benefit from progress on importance sampling in high dimensions – a concern that two of the reviewers brought up – as well as in nested optimization, where we explore an alternative approach based on hypergradients in Appendix D.

We address further individual concerns and questions in the direct responses to the respective reviewers. We hope that we have been able to further clarify the contribution of our work and the reviewers will confidently recommend acceptance.

---

> ### Author Response · Authors · 2022-08-02
> **Minor updates**
>
> Aside from changes in response to comments or questions from the reviewers, we have made the following updates to our submission:
> * We fixed a small bug in the plotting code for uncertainty visualization of the four-class toy-experiment. The corresponding figures now correctly display the entropy of the four-way classifier rather than just for one of the classes.
> * We updated the coreset points selection rule at the pruning step of Sparse Batch BBVI: Instead of selecting the top-K datapoints from the current coreset, we draw K samples from the multinomial defined by the weights of the coreset points. This represents a valid sample from the learned distribution of pointwise importances and can offer more stable inference (e.g. allowing for better coverage of all existing classes when the K largest weights belong to a subset of the classes).

---

> ### Author Response · Authors · 2022-08-02
> **Continual learning setup**
>
> Rather than observing the data as a single batch as in the paper, we start off with two initial classes and subsequently observe the remaining two classes individually. Between each step, **we discard the previous data** and only augment the coreset with additional parameters, copying previous coreset points into the new training data. We set up each successive task as a new inference problem that only carries previously learned coresets forward, rather than instantiating the full continual learning scenario here where we also add between-task regularization.
>
> Specifically, whenever a new class is observed, we make the following updates:
> - Training dataset: The training dataset used up to that moment is replaced by (i) the coreset learned so far, concatenated with (ii) the training data of the new class.
> - Coreset parameters: The variational parameters are (i) the coreset points so far, and (ii) extra coreset points initialized with labels from the new class. The total likelihood of each of these sets of points is rescaled proportionally to the corresponding dataset size.
> - Model parameters: The variational model parameters are fully reinitialised. The architecture is adapted so that the outer layer can do the (k+1)-classification. Note that this avoids the often encountered challenge of having to choose between a single- and multi-head architecture, and instead the output dimension of the network can correspond to the number of classes observed so far, which avoids optimizing parameters that are not yet in the training set.
>
> We note that of particular interest here is that we show that our method can be used to benefit from both fixed coresets (i.e. the ones trained in previous rounds) as well as an augmented set of coresets that are learned to ‘fit the gaps’ in the new data-task, conceptually achieving a similar goal to how inducing points are deployed in the VAR-GP continual learning paper.

---

### Meta-Review · Area_Chair_ZoKd · 2022-08-23

**Recommendation:** Accept
**Confidence:** Less certain

**Metareview:**

This paper bridges black-box probabilistic modeling to the use of variational 'pseudo coresets'. The reviewers seem to have reached a positive consensus and have engaged in a fruitful discussion during the rebuttal period. Overall, the contributions are solid and of interest to the community, but I recommend the reviewers take into consideration the remaining concerns raised by reviewer vKEG in preparing a thoughtful revision.

**Award:**

No

---

### Decision · Program_Chairs · 2022-09-14

Accept